# Vacancy-ordered perovskite superlattice in cerium titanate negative electrode for enhanced lithium-ion storage

Xuhui Xiong[1], Zhengwang Liu[1], Ruixuan Zhang[2], Liting Yang[1], Guisheng Liang[1], Ke Pei[1] & Renchao Che [1] ✉

Commercial negative electrodes such as graphite and $Li_4Ti_5O_{12}$ are fundamental to lithium-ion batteries but face inherent trade-offs among safety, energy density, rate performance, and cycling stability. In this work, we introduce structural ordering and vacancy engineering into a perovskite negative electrode $Ce_{2/3}TiO_3$ to tackle this dilemma, by creating highly ordered Ce vacancies that form a stable superlattice. As a result, micron-sized $Ce_{2/3}TiO_3$ achieves a high specific capacity (>200 mAh g$^{-1}$) at an optimal operating potential (~0.8 V vs. Li$^+$/Li), with fast-charging capability up to 50 C and stable cycling performance exceeding 10000 cycles at 20 C. Its electrochemical performance has the potential to overcome the shortcomings of graphite and $Li_4Ti_5O_{12}$, comparable to many representative intercalation-type negative electrodes. In situ structural analysis and atomic-scale imaging reveal a reversible topological phase transition between long-range and short-range ordering, which preserves the lattice integrity while unlocking low-barrier Li$^+$ diffusion pathways. Here, we show that vacancy ordering provides a compelling strategy for designing high-performance electrodes.

With the rapid advancement of portable electronics, electric vehicles, and large-scale energy grids, lithium-ion batteries (LIBs) have emerged as the leading contender for energy storage due to their high energy density, lightweight design, and reliability[1–3]. Next-generation LIBs are increasingly expected to operate within safe potential regimes while delivering fast-charging capability and cycling stability, essential for minimizing charging times and extending service lifespan[4,5]. This goal critically relies on the rational compositional design and precise structural engineering of electrode materials[6–8].

Graphite and $Li_4Ti_5O_{12}$ (LTO) stand as the commercially successful intercalation-type negative electrodes for LIBs, extensively employed in numerous practical applications[9,10]. Graphite offers a high specific capacity of ~ 372 mAh g$^{-1}$ and a low potential plateau of < 0.1 V (vs. Li$^+$/Li)[11,12], enabling high energy density in full cells. However, its sluggish Li$^+$ diffusion kinetics and inevitable structural degradation during

cycling severely hinder the rate capability and long-term cycling stability[13,14]. Moreover, the extremely low potential renders graphite susceptible to lithium dendrite formation, significantly increasing the risk of short circuits and thermal runaway[15]. In contrast, LTO operates at a safer potential of ~ 1.55 V (vs. Li$^+$/Li) and exhibits considerable structural stability owing to its zero-strain characteristic[16–18]. Nevertheless, its intrinsically low electrical conductivity and sluggish Li$^+$ diffusion lead to a limited specific capacity of only 50–60 mAh g$^{-1}$ at 10 C for micron-sized particles[19,20], while the relatively high potential compromises the overall energy and power densities[21]. To overcome these drawbacks, strategies such as defect engineering and nanostructuring have been widely adopted to enhance Li$^+$ transport kinetics and electronic conductivity[22–25]. For instance, Fu et al. fabricated thin LTO nanosheets that facilitated rapid Li$^+$ diffusion, achieving a specific capacity of 146 mAh g$^{-1}$ even at 100 C[26]. Similarly, Su et al. designed

[1]Laboratory of Advanced Materials, Shanghai Key Lab of Molecular Catalysis and Innovative Materials, State Key Laboratory of Coatings for Advanced Equipment, College of Smart Materials and Future Energy, Fudan University, Shanghai, China. [2]Zhejiang Laboratory, Hangzhou, China. ✉e-mail: rcche@fudan.edu.cn

LTO nanocrystals with intrinsic oxygen vacancies and cation redistribution, delivering 176 mAh g$^{-1}$ at 10 C owing to improved ionic/electronic conductivity[27]. However, such nanostructured designs often compromise volumetric energy density and cycling stability, due to increased surface reactivity, structural degradation, and pronounced volumetric effects. Thus, achieving intrinsic electrochemical performance with safety, high capacity, fast-charging capability, and cycling stability at the micrometer scale remains a critical challenge in the pursuit of advanced negative electrodes.

The degree of crystallographic ordering in electrode materials directly dictates their electrochemical performance. Structural disorder and insufficient vacancy concentration often lead to sluggish ion diffusion, electrochemically inactive behavior, and progressive structural degradation, thereby compromising capacity retention, rate capability, and long-term cycling stability[28,29]. Disorder disrupts ion transport pathways, elevates charge-transfer resistance, and accelerates undesirable phase transitions, ultimately diminishing overall electrochemical property[30]. In recent years, the concept of structural ordering design has found widespread applications in superconductivity, topological electronics, catalysis, and energy storage[31–35]. In these systems, structural ordering not only determines the intrinsic physical properties of materials but also governs the spatial arrangement of functional units, thus optimizing key performance metrics. For battery electrode materials, structural ordering plays a crucial role in facilitating ion diffusion, reducing charge-transfer resistance, and preserving structural integrity during cycling[36,37]. For example, precise control of lattice-ordering domains in Ni-rich layered positive electrodes enables efficient Li$^+$ transport and mitigates degradation from phase transitions, thus improving both high-voltage stability and cycle life[38]. Inspired by these advances, structural ordering design emerges as a powerful strategy to simultaneously enhance electrochemical activity, rate capability, and cycling stability. Nevertheless, vacancy engineering remains a critical yet frequently neglected aspect of negative electrodes, as it is crucial to maximizing Li$^+$ accommodation. Therefore, integrating structural ordering with rational vacancy engineering provides a promising route to optimize the electrochemical performance of negative electrodes.

In this context, A-site deficient perovskites with the general formula Ln$_{2/3}$TiO$_3$ (Ln = La, Ce, Pr, Nd) serve as ideal structural prototypes for intercalation-type negative electrodes, intrinsically coupling ordered cation vacancies with robust TiO$_6$ octahedral frameworks[39–43]. The vacancy ordering not only modulates the Ti-O-Ti connectivity to promote Li$^+$ transport but also mitigates volume fluctuations during cycling to enhance structural stability. Several derivatives of this family, such as Li$_{0.27}$La$_{0.54}$TiO$_{2.945}$, La$_{0.5}$Li$_{0.5}$TiO$_3$, Li$_{0.35}$Nd$_{0.55}$TiO$_3$, and Li$_{0.38}$Pr$_{0.54}$TiO$_3$, have been explored as the negative electrodes for LIBs[44–47]. However, they usually suffer from limited rate capability and insufficient long-term cycling stability, while their underlying Li$^+$ storage mechanisms at the atomic scale remain poorly understood. Therefore, exploring alternative A-site deficient perovskites is essential to overcome current performance constraints and to establish universal structure-property relationships for high-performance intercalation-type negative electrodes.

In this study, we integrate structural ordering and vacancy engineering within the perovskite negative electrode Ce$_{2/3}$TiO$_3$ (CTO), which possesses a superlattice structure with intrinsically ordered Ce vacancies (Fig. 1a). This vacancy-ordered superlattice endows CTO with great electrochemical performance in LIBs. Experimental results reveal that micron-sized CTO operates at a low yet safe potential of ~ 0.8 V (vs. Li$^+$/Li), intermediate between graphite and LTO, while delivering a higher specific capacity of 221 mAh g$^{-1}$ at 0.1 C. Furthermore, CTO exhibits high-rate capability up to 50 C and stable cycling performance exceeding 10000 cycles at 20 C, highlighting its comprehensive Li$^+$ storage performance. To elucidate the underlying mechanism, in situ X-ray diffraction (XRD) and atomic-scale imaging were employed to directly visualize and quantify the structural evolution of CTO during Li$^+$ insertion and extraction. First-principles calculations further confirm the low-barrier Li$^+$ diffusion within the vacancy-ordered superlattice, consistent with its rapid electrochemical kinetics. This study proves CTO as a promising alternative to conventional graphite and LTO negative electrodes, while providing fundamental insight into the pivotal role of vacancy ordering in electrochemical performance.

## Results

### Structural properties of vacancy-ordered perovskite superlattice CTO

As illustrated in Fig. 1b, CTO possesses a characteristic A-site deficient perovskite structure, which consists of corner-sharing TiO$_6$ octahedra interspersed with partially occupied Ce sites. These Ce sites are arranged in an orderly manner, forming alternating Ce-rich and Ce-poor layers with distinct occupancy ratios that collectively construct a vacancy-ordered superlattice. The pristine CTO was synthesized via a high-temperature solid-state route, yielding micron-sized particles with sizes ranging from 5 to 20 μm (Supplementary Fig. 1). Rietveld refinement of the XRD pattern (Fig. 1c) confirms that the synthesized CTO exhibits high crystallinity and phase purity, adopting a tetragonal structure with the *P4/mmm* space group. The refinement results (Supplementary Table 1) show that the average occupancy of the Ce1 site is 0.889, which is higher than that of the Ce2 site (0.444). Accordingly, Ce1 sites with fewer vacancies form Ce-rich layers, while Ce2 sites with more vacancies constitute Ce-poor layers. This structural configuration is consistent with the ideal model of a vacancy-ordered superlattice.

To directly visualize the vacancy-ordered superlattice structure of CTO, high-angle annular dark-field scanning transmission electron microscopy (HAADF-STEM) images with atomic resolution were acquired along the [100] zone axis. As shown in Fig. 1d, the large-scale atomic arrangement exhibits a well-defined periodic pattern with alternating bright and dark atomic columns. A magnified region aligns perfectly with the theoretical crystal structure of CTO. The brighter atomic columns correspond to Ce1 sites with higher occupancy, while the darker atoms represent Ce2 sites with a higher vacancy concentration. Smaller Ti atoms occupy positions between the Ce1 and Ce2 layers. The line intensity profile along the vertical direction further confirms the distinct contrast differences between Ce1 and Ce2 sites, consistent with the structural model of CTO. The selected area electron diffraction (SAED) pattern shows bright diffraction spots that align with the simulated results (Supplementary Fig. 2), while weak superlattice reflections in the interstitial regions indicate long-range structural modulation arising from the ordered Ce distribution. To further verify the ordered arrangement of Ce vacancies, integrated differential phase contrast scanning transmission electron microscopy (iDPC–STEM) imaging was conducted, which enhances contrasts for both heavy and light elements. As shown in Fig. 1e, the Ce-rich layers exhibit higher contrast owing to more complete Ce occupancy, whereas the Ce-poor layers display pronounced vacancy features. This alternating distribution of Ce-rich and Ce-poor layers constitutes the unique vacancy-ordered perovskite superlattice. In addition, atomic-resolution energy dispersive X-ray spectroscopy (EDX) mappings (Fig. 1f) validate the high phase purity and uniform atomic arrangement of CTO. The periodic fluctuation in Ce signal intensity suggests the different occupancy of Ce sites, and the nearly uniform Ti signal reflects the homogeneous distribution of Ti atoms. Furthermore, the electron energy loss spectroscopy (EELS) spectrum (Supplementary Fig. 3) identifies the presence of Ce$^{3+}$ and Ti$^{4+}$. Specifically, the Ti L$_{2,3}$-edge peaks at 458.1 eV and 463.2 eV correspond to Ti$^{4+}$ 2p$_{2/3}$ and 2p$_{1/2}$ transitions[48], and the Ce M$_{4,5}$-edge peaks at 880.2 eV and 897.8 eV are attributed to Ce$^{3+}$ 3d$_{5/2}$ and 3d$_{3/2}$ transitions[49,50]. Collectively, these results confirm the successful synthesis of a vacancy-ordered perovskite superlattice CTO, in great agreement with the predicted crystallographic model.

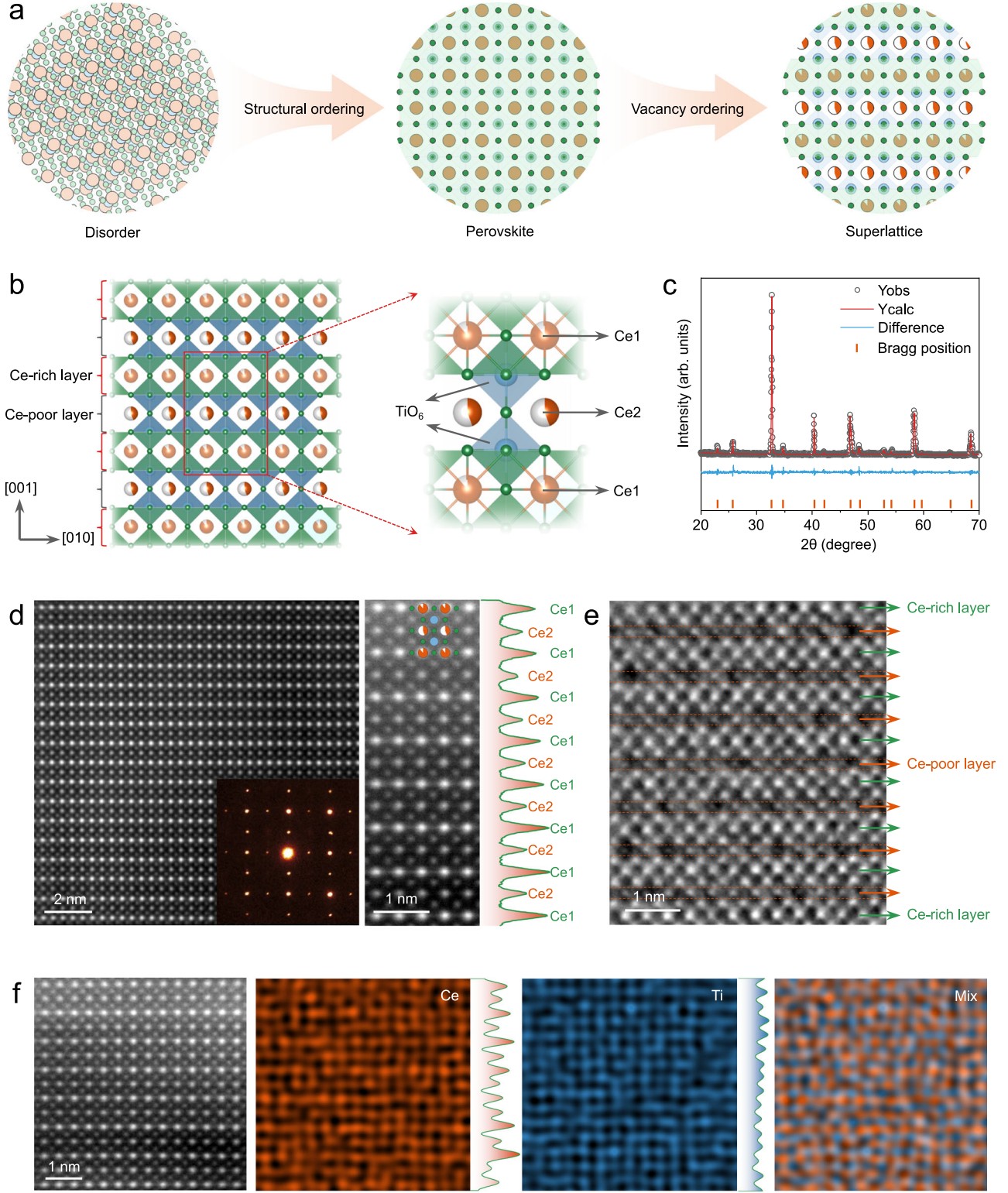

**Fig. 1 | Crystalline structure of vacancy-ordered perovskite superlattice CTO.**
**a** Schematic of structural ordering and vacancy engineering (orange, blue, and green spheres denote A-site, B-site, and oxygen atoms of the perovskite, respectively). **b** Periodic superlattice structure and unit-cell model of CTO. **c** Powder XRD pattern with Rietveld refinement. **d** HAADF-STEM image along the [100] zone axis with the corresponding SAED pattern in the inset. A magnified view and atomic-intensity profile are presented on the right (orange sphere: Ce, blue spheres: Ti, green spheres: O). **e** iDPC-STEM image along the [100] zone axis. **f** Atomic-resolution EDX mappings.

## Electrochemical Li⁺ storage performance of micron-sized CTO

Given the structural advantages of CTO in accommodating Li⁺, the electrochemical performance of the prepared micron-sized CTO was systematically evaluated in half-cell configurations. The initial discharge/charge cycle (Supplementary Fig. 4) delivered specific capacities of 343 and 221 mAh g⁻¹ for Li⁺ insertion and extraction, respectively, corresponding to an initial coulombic efficiency of 64.4%, as corroborated by the cyclic voltammetry (CV) curves in Fig. 2a. The

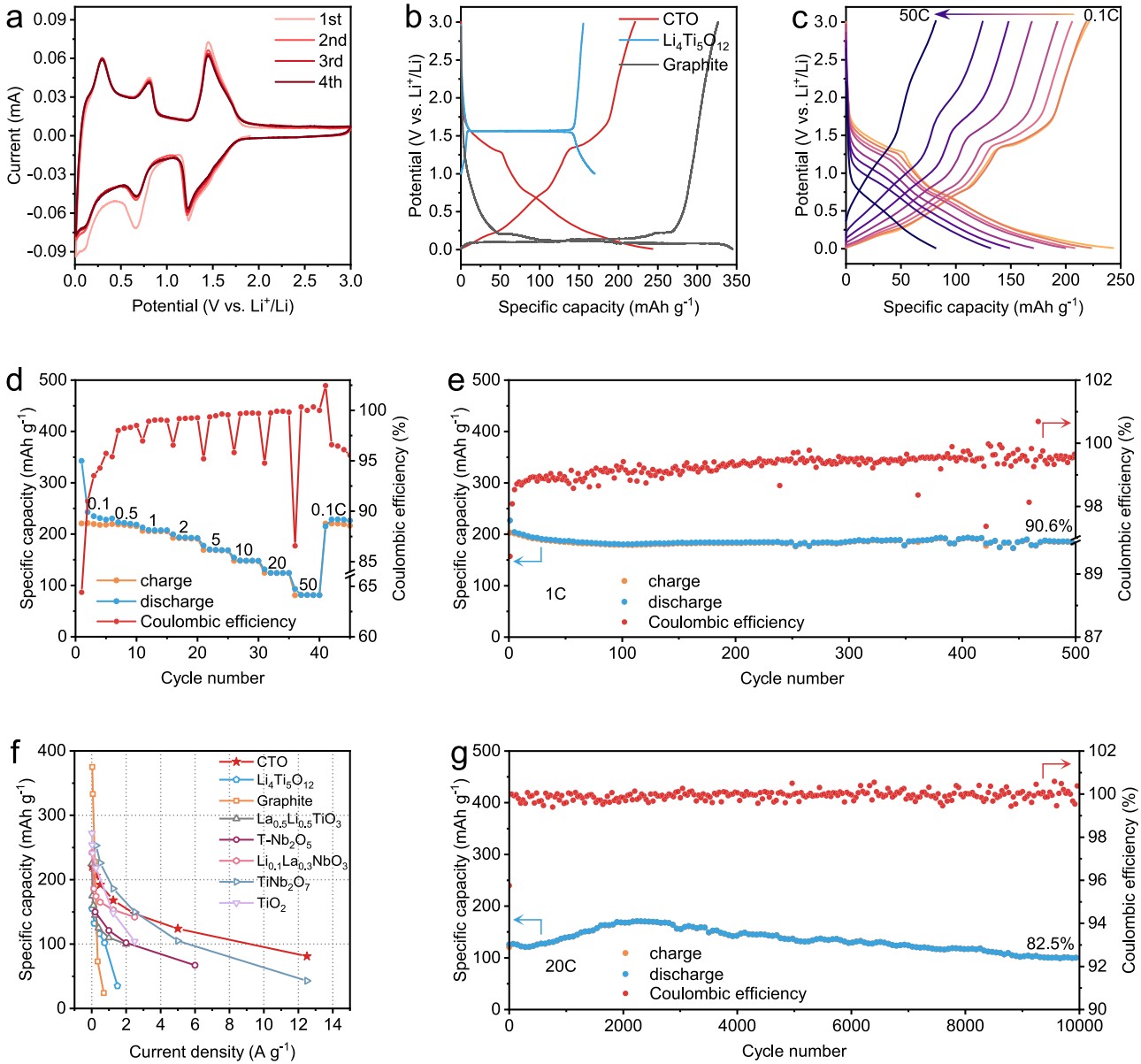

**Fig. 2 | Electrochemical performance of Li||CTO half cells. a** CV curves for the initial four charge-discharge cycles at a scan rate of 0.2 mV s$^{-1}$. **b** Charge-discharge curves of CTO, LTO, and graphite, the potential windows are 0.01–3.0 V, 1.0–3.0 V, and 0.01–3.0 V, respectively. **c** Charge-discharge profiles at various rates from 0.1 C to 50 C (1 C = 250 mA g$^{-1}$). **d** Rate capability and coulombic efficiency at 0.1, 0.5, 1, 2, 5, 10, 20, and 50 C. **e** Cycling performance at 1 C. **f** Comparison of rate performance of CTO with representative negative electrodes, with a more detailed summary presented in Supplementary Table 2. **g** Long-term cycling performance at 20 C.

initial capacity loss is primarily attributed to the formation of the solid electrolyte interphase (SEI). As shown in Supplementary Fig. 5, transmission electron microscopy (TEM) images reveal progressive thickening of the SEI layer on CTO particle surfaces during the initial discharge. High-resolution Li 1s X-ray photoelectron spectroscopy (XPS) spectra indicate that the SEI consists of both organic (ROCO$_2$Li) and inorganic (Li$_2$CO$_3$ and LiF) components, which are from different interfacial reactions. ROCO$_2$Li and Li$_2$CO$_3$ arises from the reductive decomposition of carbonate solvents and side reactions with conductive carbon, while LiF forms through the decomposition of LiPF$_6$ salt. As lithiation proceeds, the relative intensities of Li$_2$CO$_3$ and LiF signals increase while those of organic species decrease, suggesting a gradual enrichment of inorganic SEI components at low potentials and corresponding irreversible lithium consumption. After prolonged cycling, the SEI maintains a similar chemical composition but becomes slightly thicker, with further accumulation of inorganic components such as LiF and Li$_2$CO$_3$ (Supplementary Fig. 6). This dense, inorganic-

rich SEI enhances interfacial mechanical and chemical stability, effectively suppressing continuous electrolyte decomposition and enabling long-term cycling durability. Beyond the initial cycle, subsequent CV curves at 0.2 mV s$^{-1}$ exhibit nearly consistent profiles, indicating the high electrochemical reversibility of CTO. Three distinct pairs of redox peaks are observed. The pairs at 1.45 V/1.24 V and 0.81 V/0.66 V correspond to the two-step Ti$^{4+}$/Ti$^{3+}$ redox process[46], while the pair at 0.30 V/0.10 V is attributed to the Ti$^{3+}$/Ti$^{2+}$ transition during deep lithiation[51]. These assignments are corroborated by ex situ XPS analysis (Supplementary Fig. 7), where the Ce 3d spectra remain unchanged from the pristine state to 1.0 V and 0.01 V, confirming the electrochemical inertness of Ce$^{3+}$. In contrast, the Ti 2p spectra show a progressive shift of Ti$^{4+}$ peaks toward lower binding energies, indicating the stepwise reduction of Ti$^{4+}$ to Ti$^{3+}$ and subsequently to Ti$^{2+}$ upon deep lithiation.

Compared with commercial intercalation-type negative electrodes, graphite and LTO, CTO demonstrates improved operating

potential and specific capacity (Fig. 2b). Specifically, CTO operates at a safer potential of ~ 0.8 V (vs. Li$^+$/Li) than that of graphite, which effectively mitigates the risk of lithium dendrite formation. In addition, CTO delivers a higher reversible specific capacity of 220 mAh g$^{-1}$ at 0.1 C than that of LTO, thereby enabling a higher overall energy density. Moreover, CTO exhibits high-rate charge/discharge performance, maintaining high specific capacities even at rates up to 50 C (Fig. 2c). As the current rate increases from 0.1 C to 50 C, the reversible capacities gradually decrease from 220 to 218, 206, 192, 169, 148, 124, and 81 mAh g$^{-1}$, corresponding to retention ratios of 99.1, 93.6, 87.3, 76.8, 67.3, 56.4, and 36.8%, respectively. Upon returning to 0.1 C, the reversible capacity recovers to its initial value (Fig. 2d), confirming the fast-charging capability and stability of CTO. When benchmarked against intercalated-type negative electrodes such as graphite, LTO, La$_{0.5}$Li$_{0.5}$TiO$_3$, etc.[45,52–57], CTO demonstrates both high specific capacities and improved rate performance (Fig. 2f and Supplementary Table 2). In terms of cycling performance, CTO exhibits long-term durability at both low and high rates. As plotted in Fig. 2e, CTO retains a specific capacity of 184 mAh g$^{-1}$ after 500 cycles at 1 C, corresponding to a retention of 90.6%. Even at higher rates of 10 C and 20 C, the capacity retention remains above 80% after extended cycles (Fig. 2g and Supplementary Figs. 8 and 9). These results indicate the robustness of the CTO lattice and its ability to sustain long-term cycling stability, showing performance comparable to other intercalation-type negative electrodes for LIBs (Supplementary Fig. 10 and Supplementary Table 3).

Many fast-charging negative electrodes reported to data rely on thin electrodes with low mass loadings (1-2 mg cm$^{-2}$), which limit their practical applicability. To evaluate the practical potential of CTO, we examined thick electrodes with a high mass loading of ~ 10.2 mg cm$^{-2}$. As shown in Supplementary Fig. 11, the CTO electrode delivers specific capacities of 193, 173, 160, 146, 117, 89, 63, and 30 mAh g$^{-1}$ at current rates from 0.1 C to 50 C, demonstrating great rate capability even under high-loading conditions. Moreover, it retains 80% of its initial capacity after 1000 cycles at 20 C, indicating considerable cycling stability. Beyond gravimetric performance, we also assessed the volumetric capacity of CTO. At a low mass loading of 1.5 mg cm$^{-2}$, CTO exhibits a maximum volumetric capacity of 550 mAh cm$^{-3}$, and still achieves 394 mAh cm$^{-3}$ at the high loading of 10.2 mg cm$^{-2}$ (Supplementary Fig. 12). This performance is comparable to that of previously reported micron-sized or nano-structured intercalation-type negative electrodes (Supplementary Fig. 12c), underscoring the practical advantages of CTO for high-energy-density and fast-charging LIBs.

Low-temperature performance is also a critical metric for practical battery applications, so the electrochemical performance of CTO was evaluated at 0 °C and − 15 °C (Supplementary Figs. 13 and 14). At 0 °C, CTO exhibits only a moderate capacity reduction compared to 25 °C, still delivering a reversible capacity of 183 mAh g$^{-1}$ at 0.1 C. Moreover, it maintains a great rate capability, delivering 170, 161, 149, 130, 111, 89, and 54 mAh g$^{-1}$ as the rate increases from 0.5 C to 50 C. Long-term cycling tests further demonstrate durable stability, with negligible capacity decay over 1000 cycles at 10 C and over 10000 cycles at 20 C. Even at a lower temperature of − 15 °C, where Li$^+$ diffusion kinetics are further hindered, CTO retains considerable cycling stability at 5 C and 10 C. The low-temperature performance of CTO is comparable to conventional commercial negative electrodes such as LTO and graphite[58–60], highlighting its promising potential for practical applications.

Furthermore, the practical performance of CTO was evaluated in a full-cell configuration (CTO∥LiFePO$_4$). The comparison of charge/discharge curves shows that the operating potential difference between LiFePO$_4$ and CTO reaches 2.76 V (Supplementary Fig. 15), which is favorable for a high output energy density. The full cell delivers a high discharge capacity of 150 mAh g$^{-1}$ at 0.1 C, with a first-cycle coulombic efficiency of 62.5% (Supplementary Fig. 16a), comparable to

that of the half cell. Upon increasing the current rate to 0.5, 1, 2, 5, and 10 C, the full cell maintains specific capacities of 125, 115, 108, 100, and 92 mAh g$^{-1}$, respectively (Supplementary Fig. 16b). The capacity retention at 10 C reaches 61.3%, demonstrating reliable rate capability. Furthermore, long-term cycling at 5 C confirms electrochemical stability, with 80% capacity retention after 1000 cycles (Supplementary Fig. 16c). Thus, the practical potential of CTO is proven, combining its high specific capacity, fast-charging capability, and cycling stability in a full-cell configuration.

## Electrochemical kinetic analysis for Li$^+$ diffusion

To elucidate the underlying mechanisms responsible for the high-rate performance, electrochemical kinetic analysis was performed to investigate Li$^+$ diffusion behavior. As shown in Fig. 3a, the CV curves retain their characteristic shape with progressively enhanced peak currents and small polarization as the scan rate increases from 0.2 mV s$^{-1}$ to 0.4, 0.7, and 1.1 mV s$^{-1}$, indicating high electrochemical reversibility and rapid reaction kinetics. Based on the relationship between scan rate and peak current, the $b$-values were calculated to evaluate the reaction kinetics (Supplementary Note 1). The obtained $b$-values for the cathodic and anodic peaks are 0.78 and 0.77 (Fig. 3b), respectively, corresponding to Li$^+$ insertion and extraction processes. The $b$-values between 0.5 and 1.0 suggest kinetic behavior controlled by a combination of both diffusion and surface-capacitive mechanisms. To quantify these contributions, a current-scan rate separation model was employed (Supplementary Note 2). The analysis reveals high pseudocapacitive contribution ratios in the micron-sized CTO, accounting for 72, 76, 82, and 86% of the total capacity at 0.2, 0.4, 0.7, and 1.1 mV s$^{-1}$, respectively (Fig. 3c and Supplementary Fig. 17). These pseudocapacitive behaviors are comparable to those of many nanoscale electrodes[61–63], which facilitate fast charge storage and mitigate structural degradation caused by deep Li$^+$ diffusion.

The Li$^+$ diffusion coefficients in CTO were further determined through a galvanostatic intermittent titration technique (GITT). As depicted in Fig. 3d, alternating current pulses and relaxation steps were performed at a low rate of 0.1 C, aiming to probe Li$^+$ diffusion during lithiation and delithiation under near-equilibrium conditions. As shown in the enlarged view (Fig. 3e), $\triangle E_\tau$ denotes the potential change during a single current pulse, while $\triangle E_s$ represents the potential difference between two adjacent equilibrium states. Based on Fick's law of diffusion (Supplementary Note 3), the Li$^+$ diffusion coefficients were calculated as a function of potential. The average diffusion coefficients during lithiation and delithiation are $5.29 \times 10^{-11}$ and $9.39 \times 10^{-11}$ cm$^2$ s$^{-1}$, respectively, which are comparable to those of many fast-charging electrodes (Supplementary Table 4). These results confirm the rapid Li$^+$ diffusion within the micron-sized CTO structure. The Li$^+$ diffusion coefficient exhibits a sharp decline near ~ 1.25 V during both lithiation and delithiation, followed by an increase below this potential (Fig. 3f). This behavior suggests a possible phase transition, accompanied by accelerated Li$^+$ diffusion kinetics in the post-transition phase.

## In situ XRD analysis of the sequential lithiation process

To gain insight into the sequential lithiation process in CTO, in situ XRD analysis was conducted in tandem with electrochemical discharge/charge measurements. As depicted in Fig. 4a, b, the evolution of in situ XRD patterns was synchronized with the discharge/charge curves at 0.5 C, providing real-time information on structural changes. In the pristine state, CTO exhibited three distinct diffraction peaks at 25.6°, 32.6°, and 40.2°, corresponding to the (101), (110), and (112) planes of the tetragonal structure (Supplementary Fig. 18). As lithiation proceeded, these diffraction peaks gradually shifted to lower angles, reflecting lattice expansion driven by Li$^+$ insertion. A transition in the diffraction pattern occurred near 1.25 V, where the peaks shifted more rapidly toward lower angles until full lithiation was reached. No new

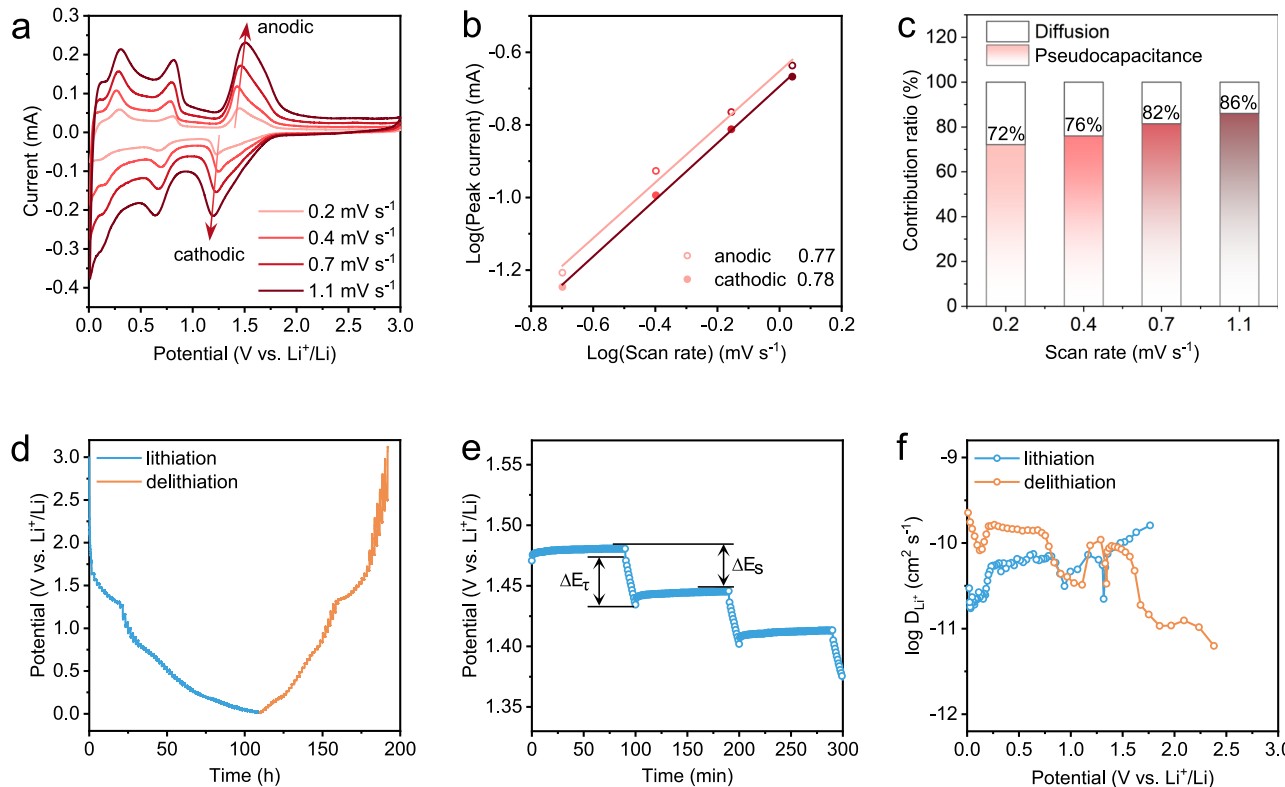

**Fig. 3 | Kinetic analysis of Li⁺ diffusion within the CTO electrode. a** CV curves at varying scan rates of 0.2, 0.4, 0.7, and 1.1 mV s⁻¹. **b** Logarithmic relationship between peak current and scan rate. **c** Quantitative assessment of pseudocapacitive contributions in CTO. **d** GITT profiles during lithiation and delithiation at 0.1 C (25 mA g⁻¹). **e** Enlarged view of GITT curves for the determination of $\triangle E_\tau$ and $\triangle E_s$. **f** Calculated Li⁺ diffusion coefficients during lithiation and delithiation.

peaks, peak splitting, or disappearance was observed, suggesting that the transformation around 1.25 V involved a topological phase transition. Upon delithiation, the diffraction peaks returned to their original positions, demonstrating the considerable structural reversibility of CTO. XRD Rietveld refinement below 1.25 V corroborated the retention of the tetragonal system, but with a reduction in the unit-cell parameters *c* that approached the value of parameter *a* (Supplementary Fig. 19). This structure was thus identified as a pseudo-cubic phase, exhibiting a similar structural symmetry to the cubic phase. Raman spectra collected across the phase transition region (Supplementary Fig. 20) reveal systematic vibrational changes during lithiation. The gradual downshift of main peaks, along with the progressive merging of low-frequency modes, indicates an increased structural symmetry, consistent with the transition toward a pseudo-cubic phase with higher symmetry. The refinement structural parameters displayed abrupt changes in the unit-cell parameters *a*, *c*, and lattice volume *V* at the transition point, confirming a topological phase transition between tetragonal and pseudo-cubic phases (Fig. 4c, d). Despite the noticeable unit-cell volume variation during lithiation, the overall periodicity of the original tetragonal structure was preserved. This topological phase transition underscores the structural resilience of CTO, which is favorable for achieving long-term cycling stability.

Moreover, the unit-cell parameters *a* and *c* displayed gradual variations during the early stage of Li⁺ insertion, followed by a rapid expansion after the transition point at 1.25 V. This indicates that the post-transition phase provides a more favorable structural environment for Li⁺ diffusion and insertion, in agreement with the calculated Li⁺ diffusion coefficients. Based on the discharge curve and the above results (Fig. 4e), we propose a possible sequence for the lithiation process. Initially, Li⁺ ions preferentially insert into the Ce-poor layers, which possess a higher vacancy concentration that facilitates Li⁺ diffusion. As the Ce-poor layers become partially filled, the CTO lattice

undergoes a topological phase transition, generating the pseudo-cubic phase. Ultimately, Li⁺ ions occupy the residual vacancies in both Ce-poor and Ce-rich layers, resulting in the complete transformation of CTO into the pseudo-cubic structure.

## Atomic-scale structural evolution during Li⁺ insertion

Due to the limitations of in situ XRD analysis in resolving topological structural information, atomic-resolution scanning transmission electron microscopy (STEM) was performed to directly visualize the structural evolution of CTO at various lithiation states. As shown in Fig. 5a–c, HAADF-STEM images were acquired along the [110] zone axis for three representative states: pristine, discharged to 1.25 V, and fully discharged to 0.01 V, corresponding to the initial, phase transition, and fully lithiated states, respectively. In the pristine state, CTO exhibits a well-defined vacancy-ordered superlattice, characterized by alternating Ce-rich and Ce-poor layers with pronounced contrast differences (Fig. 5a). Upon discharge to 1.25 V, a two-phase coexistence becomes evident (Supplementary Fig. 21), marked by different colors (Fig. 5b). As the vacancies in Ce-poor layers are partially occupied by Li⁺, the contrast difference between Ce-poor and Ce-rich layers diminishes. Consequently, a topological phase emerges, characterized by nearly uniform contrast across all Ce sites, indicative of a pseudo-cubic structure. When fully discharged to 0.01 V, Li⁺ ions occupy all available vacancies in both Ce-rich and Ce-poor layers, leading to a complete structural transformation of the superlattice into a pseudo-cubic phase (Fig. 5c). The SAED pattern along [100] zone axis after transformation shows additional weak satellite reflections surrounding the primary diffraction spots, which is different from those in the pristine state, proving the formation of a short-range ordered superlattice (Supplementary Fig. 22)[64,65]. Since the pseudo-cubic structure retains similar periodicity to the original vacancy-ordered superlattice but exhibits a halving of the lattice parameter along the *c*-axis, this

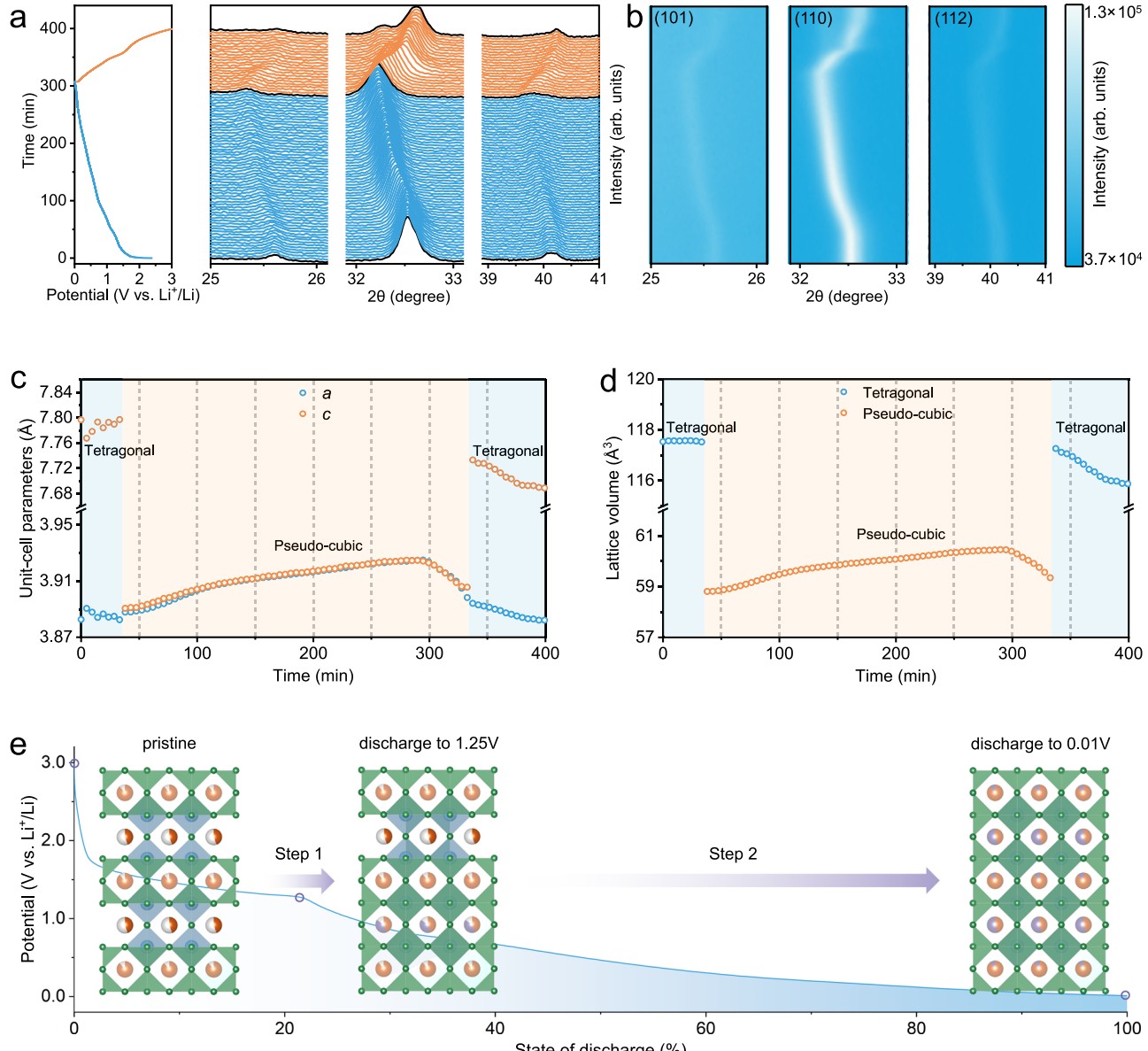

**Fig. 4 | In situ structural characterization of the CTO negative electrode during Li⁺ insertion and extraction. a** In situ XRD patterns recorded together with the discharge-charge curves during the first cycle at 0.5 C (125 mA g⁻¹) and 25 °C. **b** Pseudo-color contour map showing three representative diffraction peaks. **c** Evolution of the unit-cell parameters *a* and *c* during the discharge/charge process. **d** Variations of the unit-cell volume *V* throughout the discharge-charge cycle. **e** Correlation between Li⁺ insertion steps and the discharge curve (orange: Ce, blue: Ti, green: O, purple: Li).

structural transformation is identified as a topological phase transition. This Li⁺-induced transformation represents a shift from long-range to short-range ordering, as schematically illustrated in Fig. 5d–f. Upon charging back to 3.0 V, the structure reverts to its original vacancy-ordered superlattice (Supplementary Fig. 23), consistent with the in situ XRD analysis. Even after prolonged cycling, the CTO maintains its morphological and structural integrity, with large-scale vacancy-ordered superlattices interspersed with minor pseudo-cubic domains (Supplementary Figs. 24 and 25), demonstrating the high reversibility and stability of the topological phase transition in CTO.

The chemical states of the constituent elements in CTO at different lithiation states were probed through EELS. As shown in Supplementary Fig. 26, the Ti L₂,₃-edge signals progressively shift to lower energies during lithiation, confirming the valence reduction of Ti⁴⁺ caused by Li⁺ insertion. In contrast, the Ce M₄,₅-edge peaks remain essentially unchanged (Supplementary Fig. 27), suggesting the electrochemical stability of Ce cations. This chemical inertness of Ce helps

mitigate lattice distortions associated with the redox activity of the Ti, thereby enhancing structural robustness. To further analyze lattice strain, two-dimensional strain distribution maps were reconstructed for different lithiation states using geometric phase analysis (GPA)[66,67]. As depicted in Fig. 5g, the color scale in the strain maps and the full width at half maxima (FWHM) of the histograms represent the distribution and magnitude of lattice strain. The pristine superlattice exhibits a homogeneous and low-strain state, whereas the lithiated lattice becomes more distorted with higher strain values. Especially, the lattice strain at 1.25 V shows the highest strain heterogeneity due to the coexistence of two phases with large lattice mismatches, yet the average strain remains as low as +0.4%. The results confirm the low-strain structural characteristic of CTO, consistent with its long-term cycling stability.

To further elucidate the evolution of structural periodicity, fast Fourier transform (FFT) and inverse FFT (IFFT) analyses were performed for pristine and lithiated CTO (Fig. 5h). In the pristine state, the

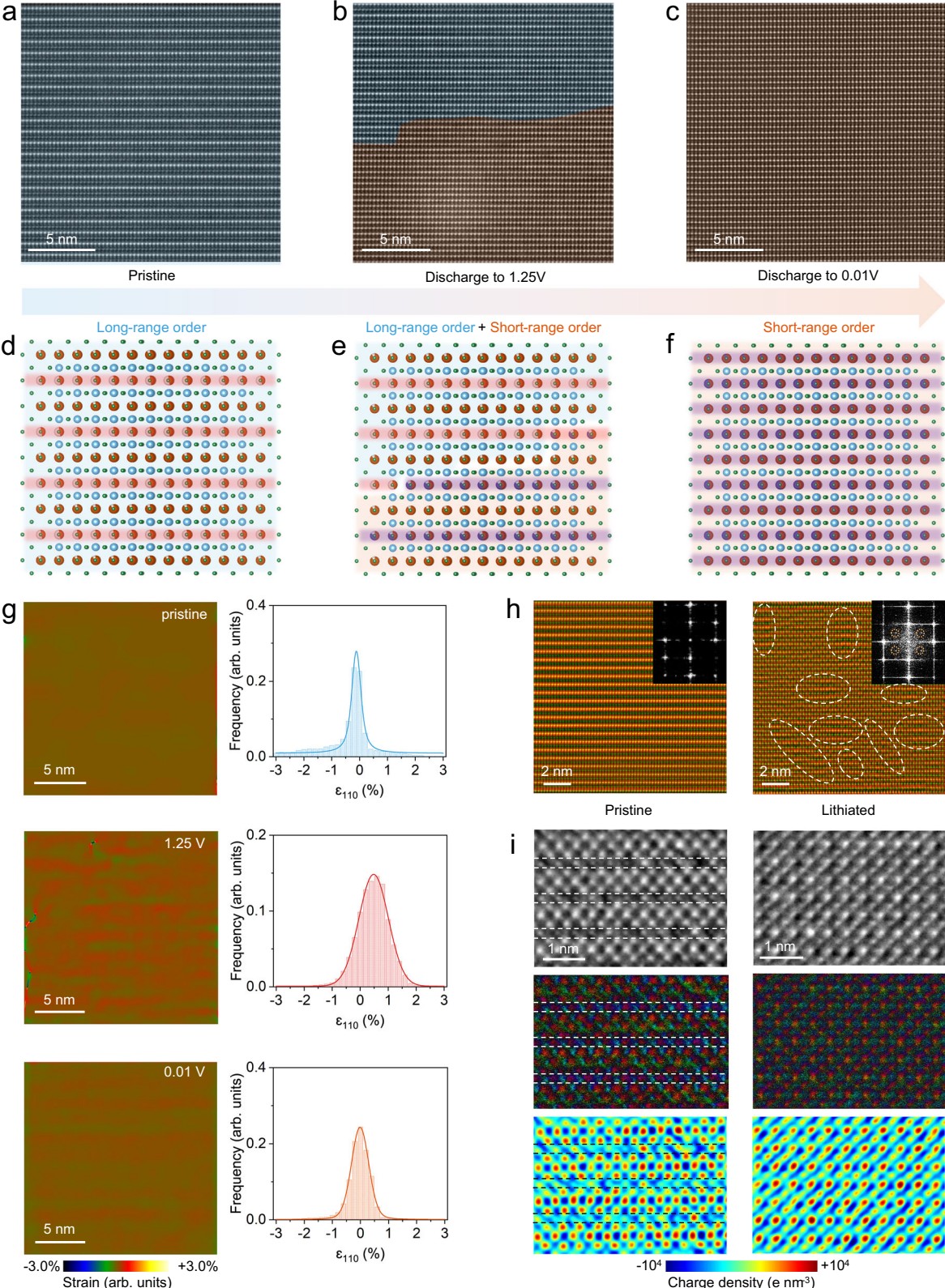

**Fig. 5 | Topological structural transformation induced by Li⁺ insertion.**
**a**–**c** HAADF-STEM images of CTO at different discharge states during the first cycle at 0.1 C (25 mA g⁻¹) and 25 °C: pristine (**a**), 1.25 V (**b**), and 0.01 V (**c**). **d**–**f** Schematics of the lithiation-induced topological structural transformation (orange: Ce, blue: Ti, green: O, purple: Li). **g** Strain distribution maps and statistical histograms of the (110) crystalline plane. **h** IFFT images of pristine and lithiated CTO derived from diffraction spots, extra diffraction spots are highlighted by yellow circles, and short-range ordered regions are marked by white circles. **i** Comparative iDPC-STEM, DPC-STEM, and charge density maps of pristine and lithiated CTO (discharged to 0.01 V).

IFFT image shows a long-range homogenous atomic arrangement, with no additional diffraction spots beyond those of CTO in the FFT pattern. In contrast, the FFT pattern of lithiated CTO reveals extra diffraction spots within the specific intervals, confirming the emergence of additional ordering. The corresponding IFFT image further proves the presence of short-range ordered domains. Similar structural modulations are observed along the [100] zone axis (Supplementary Figs. 28 and 29), originating from the topological transition to the pseudo-cubic phase. Pair distribution function (PDF) analysis was performed to compare the pristine and lithiated states. As shown in Supplementary Fig. 30a, the PDF refinements reveal a transition from long-range tetragonal ordering to short-range pseudo-cubic ordering, consistent with the structural models in Supplementary Fig. 30b. In the short-range region (3.1 - 6.1 Å), the intensities of Ce1-Ti and Ce1-Ce2 correlations increase, and a new Ce-O correlation emerges at ~ 5.0 Å upon lithiation (Supplementary Fig. 30c), indicating rearrangement of the Ce coordination environment. Moreover, the Ce1-Ce1, Ce-O, and Ce1-Ce2 peaks become noticeably narrower after lithiation (Supplementary Fig. 30d), reflecting enhanced short-range ordering of Ce atoms. These results collectively demonstrate that lithiation triggers a topological transition characterized by strengthened short-range structural ordering around the Ce coordination environment. To elucidate the effect of the transition from long-range to short-range ordering, atomic-resolution iDPC-STEM imaging were performed to derive the electric field distribution and charge density distributions (Supplementary Fig. 31)[68,69]. As shown in Fig. 5i, the pristine lattice displays alternating vacancy-rich and vacancy-poor layers, with distinct electric field and charge density distributions between these layers, as evidenced by the DPC-STEM and charge density maps. After lithiation, the lattice evolves into a uniform atomic arrangement with nearly identical electric fields and charge densities at all Ce sites. This direct atomic-level visualization of structure, electric field, and charge density provides clear evidence of a pronounced topological phase transition from long-range to short-range ordering induced by Li$^+$ insertion. These findings align well with the above analyses, offering a comprehensive understanding of the structural evolution in CTO during lithiation.

## Vacancy-ordered superlattice enabled Li$^+$ storage mechanism

To elucidate the Li$^+$ storage mechanism in CTO, density functional theory (DFT) calculations based on first principles were performed. A vacancy-ordered tetragonal superlattice model ($Ce_{10}Ti_{16}O_{48}$) was constructed to simulate the lithiation behavior, which closely approximates the experimentally observed framework of CTO. Seven types of Li$^+$ insertion sites ($Li_1$ – $Li_7$) were identified (Supplementary Fig. 32). Specifically, $Li_1$ and $Li_2$ correspond to Ce vacancies in the Ce-poor layers, while $Li_3$ resides in the Ce-rich layers. In addition, $Li_4$ – $Li_7$ occupy O4 windows between adjacent $TiO_6$ octahedra, consistent with the theoretical capacity and experimental observations (Supplementary Fig. 33). Due to the high concentration of ordered vacancies, all predicted sites exhibit low formation energies (Supplementary Table 5), suggesting the structural openness and favorable kinetics of the tetragonal superlattice for Li$^+$ accommodation. Among them, $Li_1$ presents the lowest formation energy, indicating it as the favorable initial insertion site. Once $Li_1$ is occupied, the Ce-poor and Ce-rich layers become structurally equivalent in both atomic ratios and positions, thereby enhancing the structural symmetry of the tetragonal superlattice and driving a topological phase transition toward a pseudo-cubic structure (Fig. 6a).

To investigate the lithiation behavior in the transformed phase, a partially lithiated pseudo-cubic model ($LiCe_5Ti_8O_{24}$) was constructed, in which five types of Li$^+$ insertion sites ($Li_8$ – $Li_{12}$) were identified (Supplementary Fig. 34). $Li_8$ and $Li_9$ correspond to unoccupied Ce vacancies in the Ce-poor and Ce-rich layers, respectively, while $Li_{10}$ – $Li_{12}$ occupy the O4 windows between adjacent $TiO_6$ octahedra. These insertion sites exhibit comparable formation energies (Supplementary

Table 6), with Ce-vacancy sites being energetically more favorable, confirming their priority in the lithiation sequence. These results predict a sequential lithiation process from the initial tetragonal phase to a partially lithiated pseudo-cubic phase, and eventually to the fully lithiated state (Fig. 6a), consistent with the experimental observations. Concurrently, the electronic structure undergoes evolution during lithiation, as evidenced by the projected density of states (DOS). The pristine tetragonal phase exhibits a wide bandgap of ~ 1.8 eV, with the conduction band far from the Fermi level (Fig. 6b and Supplementary Data 1–3). Upon Li$^+$ insertion, the bandgap gradually narrows, accompanied by an increase in electronic states near the Fermi level. The projected DOS of Ti $d$ orbitals (Supplementary Fig. 35) reveals electron transfer from Li to Ti 3$d$ orbitals, resulting in a progressive transition from spin-symmetric to spin-asymmetric Ti 3$d$ occupancy, indicative of the stepwise reduction of Ti$^{4+}$. Moreover, the topological phase transition induces charge redistribution. Projected charge density maps along the (100) plane (Fig. 6c) reveal that localized charge accumulation appears at former Ce-vacancy sites after phase transition, in agreement with the DPC-STEM observations (Fig. 5i). This charge reorganization enhances both electronic conductivity and Li$^+$ mobility in the pseudo-cubic phase, suggesting a strong correlation between structural evolution and electrochemical properties.

To further quantify Li$^+$ diffusion in both the vacancy-ordered tetragonal superlattice and the pseudo-cubic structure, climbing image nudged elastic band (CI-NEB) calculations were carried out. Benefiting from the open channels, bond valence sum (BVS) energy maps (Fig. 6d) show continuous Li$^+$ diffusion pathways connecting adjacent $TiO_6$ octahedra and Ce vacancies in both structures. Two dominant Li$^+$ diffusion pathways were identified, including interlayer migration between Ce-poor and Ce-rich layers, and intralayer diffusion within the Ce-poor layer. CI-NEB calculations (Fig. 6d, e) show low Li$^+$ migration barriers along the interlayer pathways for both phases, with the pseudo-cubic phase exhibiting a lower energy barrier (0.47 eV) than that of the tetragonal phase (0.62 eV). Similarly, the intralayer diffusion barrier within the Ce-poor layer is reduced in the pseudo-cubic phase (0.50 eV) compared to the tetragonal superlattice (0.64 eV) (Supplementary Figs. 36 and 37), indicating that the topological phase transition accelerates Li$^+$ diffusion kinetics. Overall, the DFT results reaffirm the advantages of the vacancy-ordered superlattice in promoting efficient Li$^+$ storage and diffusion. The lithiation-induced topological transformation to the pseudo-cubic phase further enhances Li$^+$ diffusion kinetics, thereby underpinning the great electrochemical performance of CTO, including high specific capacity, fast-charging capability, and long-term cycling stability.

## Discussion

In summary, we have developed a vacancy-ordered perovskite superlattice negative electrode CTO, characterized by a highly ordered cation-vacancy distribution. Leveraging its intrinsic vacancy ordering, micron-sized CTO has the potential to overcome the longstanding trade-offs in intercalation-type negative electrodes, achieving a well-balanced electrochemical performance that combines a moderate operating potential (-0.8 V vs. Li$^+$/Li), high specific capacity (> 200 mAh g$^{-1}$ at 0.1 C), fast charging capability (up to 50 C), and cycling stability exceeding 10000 cycles (20 C). Full-cell prototypes further validate its practical feasibility, exhibiting considerable rate performance and long-term cycling stability. The ordered structure and vacancies in CTO enhance Li$^+$ diffusion kinetics, enabling simultaneous improvements in Li$^+$ storage capacity and rate capability. Moreover, the reversible topological phase transition between long-range and short-range ordering ensures structural reversibility and stability, thereby leading to long-term cycling durability. This work sheds light on the fundamental role of vacancy ordering in enhancing electrochemical performance, paving a way for the rational design of advanced electrodes with great Li$^+$ storage properties.

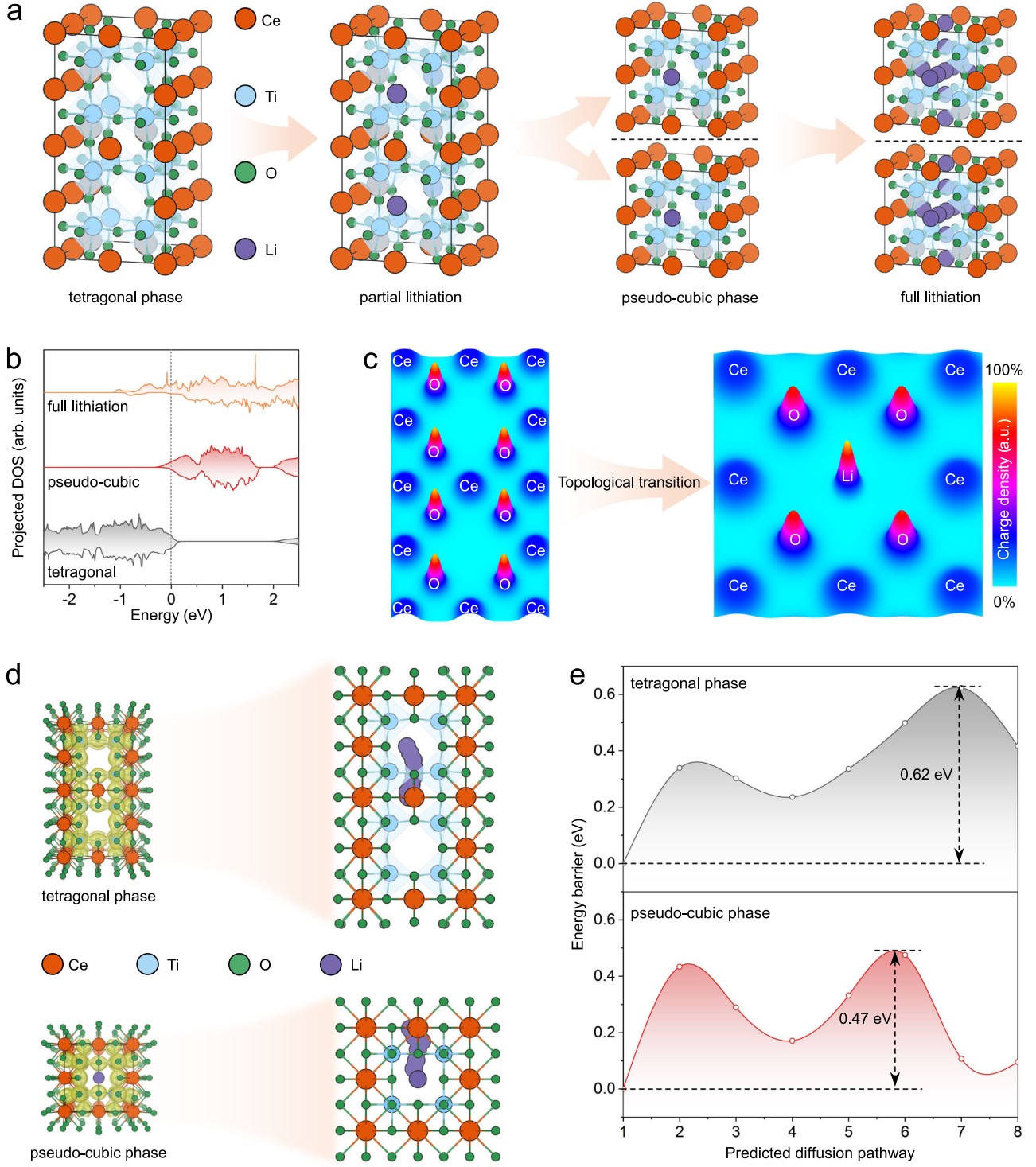

**Fig. 6 | DFT calculations of Li⁺ intercalation dynamics and phase evolution.**
**a** Geometric structures illustrating Li⁺ insertion sites and the corresponding structural evolution during lithiation. **b** Projected DOS for three representative states: the pristine tetragonal phase, the intermediate pseudo-cubic phase, and the fully lithiated phase. **c** Charge density distribution maps projected along the (100) plane for the tetragonal and pseudo-cubic phases. **d** BVS energy maps and predicted Li⁺ diffusion pathways in the tetragonal and pseudo-cubic structures. **e** Comparison of Li⁺ diffusion energy barriers in the tetragonal and pseudo-cubic structures.

## Methods

### Preparation of CTO

Micron-sized CTO was synthesized via a two-step high-temperature solid-state reaction. Initially, high-purity $CeO_2$ (≥ 99.0%, Aladdin) and $TiO_2$ (≥ 99.0%, Aladdin) powders with the stoichiometric ratio were weighed and mixed. The mixture was then subjected to high-energy ball milling (SPEX 8000 M) for 10 h. at 300 rpm using agate milling balls. Then, the resulting mixture was calcined at 1000 °C for 5 h in air to obtain a white intermediate product. In the subsequent step, the intermediate was sintered at 1200 °C for 10 h under a 5% $H_2$/Ar atmosphere, with a heating rate of 10 °C min⁻¹. After natural cooling, brownish-black CTO particles were successfully obtained.

## Structural and physical characterizations

Powder XRD data were collected using a high-resolution X-ray diffractometer (D8 Advance, Bruker), with a scan rate of 3° min$^{-1}$ and a step size of 0.01°. PDF measurements were performed on the same X-ray diffractometer equipped with an Mo-Kα radiation source ($\lambda = 0.7107$ Å) operating at 50 kV and 30 mA. The crystal structures were constructed and visualized using the VESTA software package[70]. Electrochemically synchronized in situ XRD measurements were conducted on the same X-ray diffractometer integrated with an in situ battery unit and an external electrochemical control system. Scanning electron microscopy (SEM) images were collected using a field-emission scanning electron microscope (Hitachi S-4800, Japan) operated at an accelerating voltage of 5 kV. Raman spectra were collected using a confocal Raman spectrometer (Renishaw inVia) equipped with a 633 nm excitation laser. XPS spectra were acquired on an X-ray photoelectron spectrometer (Thermo Scientific ESCALAB 250Xi). Samples were transferred to the analysis chamber using an airtight container to avoid surface contamination. The binding energies were calibrated using the C 1$s$ peak at 284.8 eV as a reference. HAADF-STEM imaging was carried out using a spherical aberration-corrected transmission electron microscope (Spectra 300, Thermo Fisher Scientific) operated at 300 kV. Atomic-resolution EDX elemental mapping was acquired by a high-resolution energy detector. EELS data were collected through a Gatan imaging filter detector, with an energy resolution of 0.8 eV for the zero-loss peak and an energy dispersion of 0.2 eV per channel. The iDPC-STEM images were generated by processing the segregated signals from four-quadrant annular detectors, while DPC-STEM images were extracted to visualize the electric-field color vectors. Color-coded atomic contrast heatmaps were reconstructed by a custom Python script, which extracted atomic column positions and intensities from the raw HAADF-STEM images. Two-dimensional lattice strain maps and statistical strain analyses were calculated based on the GPA method to quantify atomic column displacements relative to ideal positions. All STEM samples were prepared via a plasma-focused ion beam (FIB, Helios 5 PFIB DualBeam, Thermo Fisher Scientific), ensuring a specimen thickness below 100 nm. All FIB procedures were conducted following standardized protocols to ensure consistency and reproducibility. For ex situ characterizations, all cells were measured at 25 ± 1 °C and then transferred into an argon-filled glovebox (H$_2$O < 0.1 ppm, O$_2$ < 0.1 ppm, Mikrouna) for disassembly. The electrodes were sealed in airtight containers for subsequent structural and compositional analyses.

## Electrochemical measurements

Electrochemical performance was evaluated using CR2016-type coin cells, with microporous polyolefin membranes (25 μm thick, Celgard 2325, Shenzhen Kejing) serving as separators. Electrode sheets were prepared following a standard slurry-casting process. The electrode slurry consisted of active material, carbon black (Super P, Shenzhen Kejing), and polyvinylidene fluoride (PVDF, Shenzhen Kejing) in a mass ratio of 8:1:1, dispersed in N-methyl-2-pyrrolidinone (NMP, ≥ 99.0%, Sinopharm Chemical Reagent) and stirred for 24 h. The homogeneous slurry was uniformly coated on one side of copper foils (≥ 99.9%, 15 μm thick, Shenzhen Kejing) using a doctor blade. The coated electrodes were dried under vacuum at 80 °C for 10 h and subsequently punched into 12-mm diameter disks using a slicer. The average mass loadings were ~ 1.5 mg cm$^{-2}$ and ~ 10.2 mg cm$^{-2}$ for thin and thick electrodes, respectively. For half-cell configurations, commercial lithium metal foil (1.2 mm thick, 14 mm diameter, China Energy Lithium Co., Ltd.) was used as the counter electrode, while LiFePO$_4$ (≥ 99.0%, Shenzhen Kejing) electrode (active material: carbon black: PVDF = 8:1:1) with a mass loading of ~ 2.0 mg cm$^{-2}$ was employed as the positive electrode in full cells. The electrolyte (Canrd Technology Co., Ltd.) was composed of 1 M LiPF$_6$ dissolved in a 1: 1: 1 (v/v/v) mixture of dimethyl carbonate (DMC), ethylene carbonate (EC), and diethylene carbonate (DEC). The electrolyte volume used for each cell assembly was ~ 100 μL.

CV measurements were conducted using an electrochemical workstation (CHI660E, CH Instruments), while galvanostatic charge-discharge performance was assessed with a battery testing system (CT-3008, Neware). The electrochemical window for half-cells was set as 0.01-3.0 V, and a rate of 1 C was defined as a specific current of 250 mA g$^{-1}$. GITT measurements were performed at a specific current of 25 mA g$^{-1}$ with a current pulse duration of 10 min followed by a relaxation period of 90 min, with data acquired at a rate of one point per second. Electrochemical performance tests of half-cells were carried out in a constant-temperature chamber at controlled temperatures of 25 ± 1 °C, 0 ± 1 °C, and − 15 ± 1 °C. For full cells, the mass ratio of active materials in the negative and positive electrodes was optimized with an N/P ratio of ~ 1.1, and the electrochemical window was set as 0.8–3.2 V. The specific capacity of full cells was calculated based on the mass of the positive electrode. All electrochemical performance tests were conducted using at least three independent cells to ensure data reliability.

## DFT calculations

DFT calculations were conducted using the Vienna Ab initio Simulation Package (VASP) to investigate the electronic and structural properties of CTO. The generalized gradient approximation (GGA) within the Perdew-Burke-Ernzerhof (PBE) functional was employed to accurately describe the exchange-correlation interactions. To closely approximate the experimentally observed structures, a vacancy-ordered tetragonal superlattice (Ce$_{10}$Ti$_{16}$O$_{48}$) and a lithiation-induced pseudocubic phase (LiCe$_5$Ti$_8$O$_{24}$) were reconstructed. The corresponding $k$-point meshes were set to 4 × 4 × 2 and 4 × 4 × 4, respectively, to ensure adequate Brillouin zone sampling during structural relaxation. The plane-wave cutoff energy was set as 500 eV, and a force convergence criterion of 0.01 eV Å$^{-1}$ was applied to ensure robust geometric convergence. To account for the strong electron correlation effects in Ce and Ti atoms, the Hubbard U correction was incorporated into the calculations. The on-site Coulomb interaction parameters were set as 5.0 eV for Ce $f$-states and 3.0 eV for Ti $d$-states. Possible Li$^+$ insertion sites were systematically identified by calculating the formation energies of different lithiated configurations. The formation energy for each Li$^+$-inserted configuration was calculated as follows:

$$E_{formation} = E_{Li_x-pristine} - E_{pristine} - xE_{Li} \qquad (1)$$

Where $E_{Li_x-pristine}$ and $E_{pristine}$ are the total energies of the lithiated and pristine structures, respectively, $E_{Li}$ denotes the energy per atom of bulk body-centered cubic lithium, and $x$ is the number of inserted Li$^+$ ions. To further assess Li$^+$ migration kinetics, the CI-NEB method was employed to determine the transition states and migration energy barriers. The CI-NEB calculations were conducted using the same cutoff energy (500 eV) and force convergence criterion (0.01 eV Å$^{-1}$) as in geometric optimizations.

## Data availability

All data that support the findings of this study are presented in the article and Supplementary Information. Source data are provided in this paper.

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

## Acknowledgements

This work was supported by the National Natural Science Foundation of China (No. 52231007, 12327804, T2321003, 22088101, 22405050), the National Key Research Program of China (No. 2024YFA1208902, 2024YFA1408000, 2021YFA1200600), the Science and Technology Commission of Shanghai Municipality (No. 24ZR1406400), Aeronautical Science Foundation of China (No. 202400180P9001, 2024M0730P9001), Key Laboratory of High Temperature Electromagnetic Materials and Structure of MOE (No. KB202401), Fund of Science and Technology on Surface Physics and Chemistry Laboratory (No. JCKYS2023120201), and Shanghai Municipal Education Commission (No.24KXZNA06).

## Author contributions

R.C. and X.X. conceived and designed this work. X.X. carried out the synthesis, characterization, and electrochemical performance evaluation. Z.L., L.Y., and K.P. conducted structural characterization using a spherical aberration-corrected transmission electron microscope. X.X. and R.Z. performed electron micrograph analysis and DFT calculations. X.X. and G.L. conducted electrochemical measurements and structural analysis. X.X. and R.C. drafted and revised the manuscript, with Z.L., L.Y., and G.L. contributing to further refinement through discussions. All authors participated in data interpretation and the final version of the work.

## Competing interests

The authors declare no competing interests.
