## [Transparent Peer Review file · Nature Communications]

Vacancy-ordered perovskite superlattice in cerium titanate negative electrode for enhanced lithium-ion storage

Corresponding Author: Professor Renchao Che

Version 0:

Reviewer comments:

Reviewer #1

(Remarks to the Author)

This manuscript reports a new anode material for lithium-ion batteries with a perovskite structure. The electrochemical performance, kinetic properties, and energy storage mechanism were investigated in detail. A dynamic transformation between long-range and short-range ordering was claimed to contribute to the superior lithium storage performance. However, the correlation between such structural transformation and the lithium storage properties is not unequivocally elucidated. Moreover, as the full cells show low Coulombic efficiency, the practical application potential of the anode is largely reduced. On top of that, the scientific findings in this manuscript are not enough to meet the high standards of Nature Communications. Below are some specific points:

1. In Figure 5b, the boundary between two phases is not clear.
2. Apart from electron microscopy, which only shows localized information, more experimental evidence should be provided to highlight the properties of such short-range ordering.
3. What is the driving force behind the transformation between long-range and short-range ordering?
4. The theoretical calculation part clearly only considered the ion storage in the long-range ordered crystal structure. The influence of the short-range ordering is overlooked. Moreover, the formation energy of the three states does not show much difference.

Regarding the electrochemical properties:

5. Please specify the redox reactions in the CV curves.
6. The specific capacity exhibits severe fluctuation at high current rates, please provide reasonable explanation for this.
7. In Figure 3e, it is quite obvious that the potential has not reached an equilibrium state. Therefore, the accuracy of the diffusion coefficient calculation is questionable.

Reviewer #2

(Remarks to the Author)

Reviewer #3

(Remarks to the Author)

In this work, the authors report an excellent electrochemical Li⁺ storage performance using the vacancy-ordered perovskite Ce_{2/3}TiO₃ as the anode materials for half and full cells at ambient temperature, compared to other perovskite and intercalation-type anodes. Especially, the moderate operating voltage (~0.8 V) avoids the safety hazard of graphite due to generating lithium dendrites. Moreover, the authors perform an exhaustive characterization for the crystal structure and charge-discharge process. The results are interesting and valuable. However, there exist some critical problems in the current version, especially about the charge-discharge mechanism. Therefore, I can't recommend its publication on Nature Communications unless the following issues are satisfactorily resolved.

1. A-vacancy Ln_{2/3}TiO₃ is a type of perovskites that have been widely researched in other fields (Journal of Materials Science Letters, 1999, 18, 1181; Journal of Solid State Chemistry, 2000, 149, 354; J. Am. Ceram. Soc., 2002, 85 1339;

Ceramics International, 2016, 42, 1698), including $\text{Ce}_{2/3}\text{TiO}_3$ (Journal of Alloys and Compounds, 2000, 305, 72). The authors should introduce the characters of $\text{Ln}_{2/3}\text{TiO}_3$ and the considerations that $\text{Ce}_{2/3}\text{TiO}_3$ is selected in this work.

2. The authors emphasize the superiority of CTO with respect to the commercial anodes. Nowadays, a good low-temperature performance has become an urgent demand for commercial secondary batteries. In order to better satisfy the practical requirement, the low-temperature performance of CTO should be examined.

3. The authors state that the intercalated Li^+ occupy the A-site vacancies of CTO in the whole lithiation process. However, I cannot fully agree with this view. $\text{Ce}_{2/3}\text{TiO}_3$ has the A-vacancy concentration of only 1/3. The resultant theoretical capacity is only ~47 mAh/g when all A vacancies are filled. Actually, it has been reported that the inserted Li^+ may occupy the O4 window positions in perovskite anodes (Chem Mater 2013, 25, 1607). If this case holds true in this work, it should be emphatically considered in the related experimental characterization and theoretical analysis.

4. In the pristine CTO sample, $\text{Ce}_{0.889}$ and $\text{Ce}_{0.444}$ layers uniformly alternate. And, the authors conclude that the intercalated Li^+ preferentially occupy the A-vacancies in the $\text{Ce}_{0.444}$ layers. So, the distribution of inserted Li^+ should be relatively uniform in all $\text{Ce}_{0.444}$ layers. And the lithiation-induced pseudo-cubic phase should homogeneously mixes with the pristine tetragonal phase, as observed in Fig. 5g (middle 1.25 V). Why does a distinct boundary between the pure tetragonal and pure pseudo-cubic phases occur, as shown in Fig. 5b and Fig. S15? Moreover, the difference between blue and orange regions in Fig. S15 is hardly distinguished, how to determine the phase boundary?

5. The authors state that the valence configurations of Ti atoms are $3d^04s^0$ in pristine structure, and $3d^14s^0$ in lithiated structures. If so, why are remarkably asymmetric for the spin-up and spin-down PDOS of Ti-d and Ti-p in pristine structure (Fig. S22)? And why are symmetric or quasi-symmetric for the spin-resolved PDOS of Ti $3d^14s^0$ under lithiations?

6. An obviously contradictory result. The DOS results of pristine model are inconsistent in Fig. 6b and Fig. 22. For example, the former shows a wide bandgap about 2 eV, whereas the latter shows the nature of a half metal.

7. The constructed pristine CTO model is $\text{Ce}_3\text{Ti}_8\text{O}_{24}$ in Fig. 6a, not the $\text{Ce}_4\text{Ti}_8\text{O}_{24}$ described in DFT calculations, Methods. If the constructed supercell model is $\text{Ce}_3\text{Ti}_8\text{O}_{24}$, the A-vacancy concentration is up to 5/8, much higher than the experimental 1/3 in $\text{Ce}_{2/3}\text{TiO}_3$. In addition, the constructed pristine model is a quasi-cubic supercell, different from the synthesized tetragonal cell ($c \approx 2a$). How do authors ensure the reliability of DFT calculations?

8. In Fig S8, the full cell has a coulombic efficiency only ~30% for the first cycle, much lower than the 64.4% of half cell, why?

9. Line 313, "the valance reduction" may be "the valence reduction".

Reviewer #4

(Remarks to the Author)

Review:

In this manuscript, vacancy-ordered $\text{Ce}_{2/3}\text{TiO}_3$ (CTO) is reported as a perovskite-type anode with high gravimetric capacity ($> 200 \text{ mAh g}^{-1}$ at $\approx 0.8 \text{ V}$), ultrafast-rate capability ($\approx 80 \text{ mA h g}^{-1}$ at 50 C), and sustained cycling stability (90% capacity retention after 10,000 cycles). Accelerating Li^+ transport by A-site vacancy ordering would be a new route which, if confirmed under electrode loadings, can be much better than graphite and spinel $\text{Li}_4\text{Ti}_5\text{O}_{12}$. The proof of concept is achieved utilizing significant measurements, however need to refine some sections as: electrochemical access is restricted to thin ($\approx 1.5 \text{ mg cm}^{-2}$) electrodes without any volumetric benchmarking; the 10,000 cycle claim rests simply upon a single end-point value; and the "pseudo-cubic" lithiated phase, as well as potential Ce-valence changes. Consequently, I recommend minor revision with an emphasis of the need for extensive electrode level validation, detailed cycle dataset, quantitative structural analysis and direct comparison with the industry benchmarks.

First, the reversible capacity of $220\text{--}221 \text{ mA h g}^{-1}$ at 0.1 C surpasses the $\text{Ti}^{4+} \rightarrow \text{Ti}^{3+}$ theoretical limit for CTO ($\approx 142 \text{ mA h g}^{-1}$), while EELS shows no Ce valence shift, implying that the extra $\approx 80 \text{ mA h g}^{-1}$ likely arises from over-estimated active mass or parasitic side reactions rather than intrinsic redox. Second, the initial Coulombic efficiency is only 64 %, yet the solid-electrolyte interphase is uncharacterised, leaving the source of irreversible capacity loss- and its consequences for long-term stability—unclear. Third, the purported reversible transition from tetragonal to pseudo-cubic symmetry is supported only by in-situ XRD peak shifts and qualitative STEM images; rigorous Rietveld refinement of the lithiated phase and Ce-valence analysis are required to validate this structural evolution. Finally, complementary diagnostics—such as operando or ex-situ Raman spectroscopy near the 1.25 V transition and post-cycling XPS or FTIR of the electrodes- are needed to corroborate the symmetry change and identify SEI constituents responsible for the low first-cycle efficiency.

Version 1:

Reviewer comments:

Reviewer #1

(Remarks to the Author)

The authors have addressed the questions in the revised manuscript. It is thus recommended to be accepted.

Reviewer #2

(Remarks to the Author)

I co-reviewed this manuscript with one of the reviewers who provided the listed reports. This is part of the Nature Communications initiative to facilitate training in peer review and to provide appropriate recognition for Early Career

Researchers who co-review manuscripts.

Reviewer #3

(Remarks to the Author)

The authors have provided thorough and detailed responses and revisions to my comments, which appear to have addressed all of my concerns. One minor suggestion is that the solid circles representing capacity and Coulombic efficiency in all cycling plots should be clearly labeled. Otherwise, it would be difficult for beginners in battery research to distinguish which corresponds to capacity and which to Coulombic efficiency. Overall, I recommend the publication of this paper on Nature Communications.

Reviewer #4

(Remarks to the Author)

I have reviewed the manuscript and I find that the authors have complied with all comments and criticisms outlined in my earlier review of the manuscript. The revised manuscript is therefore acceptable as it is.

Response Letter to Reviewers

Responses to the comments from Reviewer 1

General comments: This manuscript reports a new anode material for lithium-ion batteries with a perovskite structure. The electrochemical performance, kinetic properties, and energy storage mechanism were investigated in detail. A dynamic transformation between long-range and short-range ordering was claimed to contribute to the superior lithium storage performance. However, the correlation between such structural transformation and the lithium storage properties is not unequivocally elucidated. Moreover, as the full cells show low Coulombic efficiency, the practical application potential of the anode is largely reduced. On top of that, the scientific findings in this manuscript are not enough to meet the high standards of Nature Communications.

Response: We sincerely thank the reviewer's careful evaluation and constructive feedback. We appreciate the positive remarks on our comprehensive investigation of the electrochemical performance, kinetics, and energy storage mechanism of CTO. We also acknowledge the reviewer's concerns and have addressed the specific points as follows:

(1) Correlation between structural transformation and Li storage properties:

In response, we have significantly reinforced this correlation through both experimental and theoretical evidence. Experimentally, we supplemented pair distribution function (PDF) analysis of pristine and lithiated CTO. The results (now presented in **Supplementary Fig. 29**) further reveal a clear transformation from a long-range ordered tetragonal phase to a short-range ordered pseudo-cubic structure upon lithiation. These results are detailed in our response to **Comment 2**. Theoretically, we have rebuilt our DFT models to reflect both the vacancy-ordered tetragonal phase and the lithiation-induced pseudo-cubic phase. By systematically comparing lithium-insertion sites, electronic structures and Li^+ diffusion pathways of the two phases, we demonstrate that the structural transformation not only enhances electronic activity but also facilitates more favorable Li^+ transport kinetics. The revised DFT results are

included in revised **Fig. 6** and **Supplementary Figs. 31–36**.

Collectively, these experimental and computational results establish a coherent mechanism whereby lithium insertion modulates lattice symmetry and electronic structure, thereby improving lithium storage and diffusion kinetics.

(2) Low first-cycle coulombic efficiency of full cells:

Through detailed analysis, we identified that this issue primarily stemmed from the suboptimal N/P ratio design in previous full-cell configurations. We sincerely apologize for this oversight and have optimized the capacity balance (N/P = 1.1-1.2). The optimized full cells show a significantly improved first-cycle coulombic efficiency of 62.5%, approaching that of the half cells (64.4%). Furthermore, the rate capability and cycling performance of the optimized full cells were greatly enhanced. It delivered a capacity retention of 61.3% at 10 C and maintained 80% of its initial capacity after 1000 cycles at 5 C, demonstrating the excellent electrochemical performance of the full cells and the practical application potential of CTO. The updated results are shown in **Supplementary Fig. 15**, and the relevant discussion has been added to the revised manuscript as follows:

Lines 255-262, Page 10: At 0.1 C, the full cell delivers a high discharge capacity of 150 mAh g⁻¹, with a first-cycle coulombic efficiency of 62.5% (**Supplementary Fig. 15a**), comparable to that of the half cell. As the rate increases to 0.5, 1, 2, 5, and 10 C, the full cell maintains specific capacities of 125, 115, 108, 100, and 92 mAh g⁻¹, respectively (**Supplementary Fig. 15b**). Notably, the capacity retention at 10 C reaches 61.3%, demonstrating excellent rate capability. Furthermore, long-term cycling at 5 C confirms outstanding stability, with 80% capacity retention after 1000 cycles (**Supplementary Fig. 15c**).

Supplementary Fig. 15 Electrochemical performance of LiFePO₄||CTO full cells. **a** Charge/discharge curves of the first cycle, showing an initial coulombic efficiency of 62.5%. **b** Charge/discharge profiles at different current rates from 0.1 C to 10 C. **c** Rate capability and long-term cycling performance at 5 C.

(3) Scientific findings and novelty:

To the best of our knowledge, this study represents the first report of a vacancy-ordered perovskite anode CTO that simultaneously achieves a safe voltage (~0.8 V), high reversible capacity (>200 mAh g⁻¹), ultrafast rate capability (up to 50 C), and exceptional cycling stability (10000 cycles). Moreover, we uncover a novel mechanism about the topological phase transition between long-range and short-range structural ordering. The electrochemical performance of CTO is outstanding compared to many state-of-the-art intercalation-type anodes, and the mechanism has not been explored in prior studies. Additionally, we have conducted full-cell validation, detailed theoretical

calculations, and advanced atomic-scale characterizations to support these findings. We believe that this work provides new insights and design paradigm for the development of high-performance intercalation-type anode materials.

We hope that the revisions and the point-by-point responses below satisfactorily address the reviewer's concerns. Thank you again for your insightful comments.

Comment 1: In Figure 5b, the boundary between two phases is not clear.

Response: We thank the reviewer for raising this important point. The phase boundary between the tetragonal and pseudo-cubic phases is primarily distinguished by the contrast difference between Ce1 (Ce-rich) and Ce2 (Ce-poor) layers in the HAADF-STEM image. In the tetragonal phase, these alternating layers show a distinct contrast difference, whereas this contrast becomes nearly uniform in the pseudo-cubic phase.

To more clearly visualize the phase boundary, we developed a Python-based image processing algorithm to extract atomic column intensities from the raw HAADF-STEM image and reconstruct a color-coded atomic contrast heatmap. This approach enhances atomic contrast visualization and enables clear identification of the phase boundary.

In response to the reviewer's comment, we have added the original HAADF-STEM image and the corresponding atomic contrast heatmap to **Supplementary Fig. 20**.

Supplementary Fig. 20 **a** HAADF-STEM image at 1.25 V. **b** Reconstructed atomic contrast heatmap derived from the raw HAADF-STEM image. The phase boundary between the tetragonal and pseudo-cubic phases is marked by white dash lines.

Comment 2: Apart from electron microscopy, which only shows localized information, more experimental evidence should be provided to highlight the properties of such short-range ordering.

Response: We appreciate the reviewer's valuable suggestion. To further substantiate the short-range ordering, we have supplemented pair distribution function (PDF) analysis on both pristine and lithiated CTO samples, as PDF provides statistically averaged structural information beyond localized electron microscopy.

The PDF refinement results show that the pristine structure is well-fitted by a long-range ordered tetragonal model, while the lithiated state is matched with a short-range ordered pseudo-cubic structure, consistent with the phase evolution observed in HAADF-STEM images. In the short-range region (3.1 - 6.1 Å), the lithiated sample exhibits enhanced peak intensities corresponding to Ce1-Ti and Ce1-Ce2 correlations, along with the emergence of a new Ce-O peak at ~5.0 Å, indicating a rearrangement of the Ce coordination environment upon lithiation. Moreover, the Ce1-Ce1, Ce-O, and Ce1-Ce2 peaks show notably narrower widths after lithiation, suggesting enhanced short-range ordering of Ce atoms in the lithiated phase. Together with the electron microscopy results, these findings provide robust evidence for the formation of a well-defined short-range ordered structure induced by lithium insertion.

The PDF results have been included in **Supplementary Fig. 29**, and the corresponding discussion has been added to the revised manuscript as follows:

Lines 402-414, Page 15: Pair distribution function (PDF) analysis was performed to compare the pristine and lithiated states. As shown in **Supplementary Fig. 29a**, the PDF refinements reveal a transition from long-range tetragonal ordering to short-range pseudo-cubic ordering, consistent with the structural models in **Supplementary Fig. 29b**. In the short-range region (3.1 - 6.1 Å), the intensities of Ce1-Ti and Ce1-Ce2 correlations increase, and a new Ce-O correlation emerges at ~5.0 Å upon lithiation (**Supplementary Fig. 29c**), indicating a rearrangement of the Ce coordination environment. Moreover, the Ce1-Ce1, Ce-O, and Ce1-Ce2 peaks become noticeably narrower after lithiation (**Supplementary Fig. 29d**), reflecting enhanced short-range

ordering of Ce atoms. These results collectively demonstrate that lithiation triggers a topological transition characterized by enhanced short-range structural ordering around Ce coordination environment.

Supplementary Fig. 29 PDF analysis revealing lithiation-induced transformation from long-range tetragonal ordering to short-range pseudo-cubic ordering. **a** PDF refinements of pristine and lithiated CTO. **b** Structural models of the long-range ordered tetragonal and short-range ordered pseudo-cubic phases. **c** Comparison of PDF peaks in the short-range region (3.1 - 6.1 \AA), where the three dominant peaks are assigned to Ce1-Ti1 (~ 3.4 \AA), Ce1-Ce1 (~ 3.8 \AA), and Ce1-Ce2 (~ 5.5 \AA) correlations, additional minor peaks are attributed to Ti-O and Ce-O distances. **d** Comparison of Ce1-Ce1, Ce-O, and Ce1-Ce2 peaks.

Comment 3: What is the driving force behind the transformation between long-range and short-range ordering?

Response: We appreciate the reviewer's insightful question. Although direct investigations on the driving force for structural transformation in CTO are limited, analogous system $\text{La}_{2/3-x}\text{Li}_x\text{TiO}_3$ (LLTO) with similar vacancy-ordered perovskite

frameworks have been extensively studied and provide valuable insight. In LLTO, it has been well established that structural symmetry and ordering are strongly modulated by Li content and A-site vacancy concentration. Specifically, prior studies (e.g., *Solid State Ionics*, 2000, 134, 219-228; *Ionics*, 2005, 11, 333-342; *Angew. Chem. Int. Ed.*, 2000, 39, 619-621; *Chem. Mater.*, 2005, 17, 2404 – 2412; *Chem. Mater.*, 2024, 36, 1197 – 1213) have shown that low Li content with high La vacancy concentration stabilizes a long-range ordered orthorhombic structure, while increasing Li content and reducing La vacancy concentration drive a transition to higher-symmetry tetragonal and pseudo-cubic structures.

By analogy, our DFT calculations reveal that Li^+ ions preferentially occupy A-site vacancies within the Ce-poor layers during early lithiation. This occupancy reduces the atomic ratio difference between Ce-rich and Ce-poor layers, thereby leading to a symmetry change from tetragonal to pseudo-cubic, along with a transition from long-range to short-range ordering. The inserted Li^+ compensates for the A-site vacancies in the tetragonal superlattice and disrupts the long-range periodicity of Ce vacancies, resulting in a short-range ordered structure. We also calculated the Gibbs free energy change (ΔG) of the transition from tetragonal to pseudo-cubic structure, which shows a negative ΔG of -1.23 eV (as shown in **Fig. R1**), indicating that the transformation is thermodynamically favorable.

Similar lithiation-induced transition from tetragonal to pseudo-cubic phase has also been reported in other perovskite oxides, such as $\text{La}_{0.5}\text{Li}_{0.5}\text{TiO}_3$ (*Nat. Commun.*, 2020, 11, 3490) and $\text{Li}_{0.38}\text{Pr}_{0.54}\text{TiO}_3$ (*Chem. Eng. J.*, 2024, 479, 147765), where the transformation was likewise attributed to Li^+ occupation of A-site vacancies and the resulting increase in lattice symmetry.

Therefore, we think that the driving force behind the transition from long-range to short-range ordering in CTO arises from lithium insertion-induced reduction of A-site vacancy concentration and the associated enhancement of lattice symmetry.

Fig. R1: Calculated Gibbs free energy change (ΔG) for the structural transformation from tetragonal to pseudo-cubic phase.

Comment 4: The theoretical calculation part clearly only considered the ion storage in the long-range ordered crystal structure. The influence of the short-range ordering is overlooked. Moreover, the formation energy of the three states does not show much difference.

Response: We thank the reviewer for this insightful comment. In the revised manuscript, we have substantially updated the theoretical calculations to more accurately reflect the lithiation behavior in both long-range and short-range ordered phases.

Specifically, we reconstructed two representative structural models: (1) a vacancy-ordered tetragonal superlattice $\text{Ce}_{10}\text{Ti}_{16}\text{O}_{48}$, corresponding to the pristine long-range ordered phase, and (2) a partial lithiation-induced pseudo-cubic structure $\text{LiCe}_5\text{Ti}_8\text{O}_{24}$, representing the short-range ordered phase. We systematically calculated the Li^+ insertion energies at various crystallographic sites in both models (**Supplementary Figs. 31 and 33**). The revised calculations also show that the formation energies for different insertion sites within the same phase are comparable (**Supplementary Tables 3 and 4**), suggesting that multiple sites are energetically favorable for lithium accommodation. This result indicates that the open framework and abundant vacancies in both phases provide numerous low-barrier sites for Li^+ insertion, thereby enabling high lithium storage capacity and fast diffusion kinetics. These findings align well with

the experimentally observed high specific capacity and excellent rate capability of CTO.

We have updated **Fig. 6** and the corresponding discussion (**Lines 428-485, Pages 16-18**) in the revised manuscript.

Supplementary Fig. 31 Optimized structure models depicting potential Li^+ storage sites in the tetragonal phase.

Supplementary Fig. 33 Optimized structure models depicting potential Li^+ storage

sites in the pseudo-cubic phase.

Supplementary Table 3 Formation energies of different Li⁺ storage sites in tetragonal phase.

Li ⁺ storage sites	Free energy (eV)	Formation energy (eV)
Pristine	-605.49906117	0.00000000
Li ₁	-611.85515242	-4.45219713
Li ₂	-611.36189135	-3.95893606
Li ₃	-611.77873128	-4.37577599
Li ₄	-611.82169986	-4.41874457
Li ₅	-611.44550103	-4.04254574
Li ₆	-611.75829348	-4.35533819
Li ₇	-611.71583378	-4.31287849

Supplementary Table 4 Formation energies of different Li⁺ storage sites in pseudo-cubic phase.

Li ⁺ storage sites	Free energy (eV)	Formation energy (eV)
Pristine	-308.48216030	0.00000000
Li ₈	-311.98629707	-1.60024265
Li ₉	-312.08157747	-1.69552305
Li ₁₀	-311.92746288	-1.54140846
Li ₁₁	-311.88556822	-1.49951380
Li ₁₂	-311.78662127	-1.40056685

Comment 5: Please specify the redox reactions in the CV curves.

Response: We sincerely thank the reviewer for this valuable suggestion. In response,

we have carefully analyzed the CV curves in **Fig. 2a** and now provide a detailed assignment of the observed redox features.

As shown in **Fig. 2a**, three distinct redox couples are identified at approximately 1.45 V/1.24 V, 0.81 V/0.66 V, and 0.30 V/0.10 V. The redox couples at 1.45 V/1.24 V and 0.81 V/0.66 V correspond to a two-step $\text{Ti}^{4+}/\text{Ti}^{3+}$ redox process, consistent with previous reports on Ti-based anodes (e.g., *Nano Energy*, 2024, 119, 109065; *Chem. Eng. J.*, 2024, 479, 147765; *Electrochem. Commun.*, 2013, 32, 5-8; *Adv. Energy Mater.*, 2023, 13, 2302015). The redox couple at 0.30 V/0.10 V is attributed to the $\text{Ti}^{3+}/\text{Ti}^{2+}$ redox transition, which usually occurs under deep lithiation below 0.5 V, as reported in other Ti-based anodes (e.g., *Energy Environ. Sci.*, 2017, 10, 1456-1464; *Adv. Sci.*, 2025, 12, 2410543; *Nano Energy*, 2022, 94, 106972; *ACS Appl. Mater. Interfaces*, 2024, 16, 898 – 906).

This redox assignment is further supported by the evolution of the Ti $L_{2,3}$ -edge EELS spectra (**Supplementary Fig. 25**), which exhibit progressive shifts of the Ti L_2 and L_3 peaks to lower energies upon discharge, indicating the stepwise reduction of Ti^{4+} toward lower valence states. Additionally, the supplemented *ex situ* XPS measurements (**Supplementary Fig. 6**) confirm this evolution, showing a sequential reduction of Ti^{4+} to Ti^{3+} and finally to Ti^{2+} during lithiation.

We have added the relevant discussion to the revised manuscript as follows:

Lines 188-196, Pages 7-8: In addition, the CV curves exhibit three distinct pairs of redox peaks. The pairs located at 1.45 V/1.24 V and 0.81 V/0.66 V correspond to the two-step $\text{Ti}^{4+}/\text{Ti}^{3+}$ redox process⁴⁶, while the pair at 0.30 V/0.10 V is attributed to the $\text{Ti}^{3+}/\text{Ti}^{2+}$ transition during deep lithiation⁵¹. These assignments are further supported by *ex situ* XPS analysis (**Supplementary Fig. 6**). It reveals that the Ce $3d$ spectra remain unchanged from the pristine state to 1.0 V and 0.01 V, confirming the electrochemical inactivity of Ce. In contrast, the Ti $2p$ spectra show a progressive shift of Ti^{4+} signal peaks toward lower binding energies, indicating the step reduction of Ti^{4+} to Ti^{3+} and subsequently to Ti^{2+} upon deep lithiation.

References:

46. Liu H., *et al.* Cation-deficient perovskite $\text{Li}_{0.35}\text{Nd}_{0.55}\text{TiO}_3$ as a high-performance

anode for lithium-ion batteries. *Nano Energy* **119**, 109065 (2024).

51. Jang M., *et al.* Two Steps Li Ion Storage Mechanism in Ruddlesden – Popper $\text{Li}_2\text{La}_2\text{Ti}_3\text{O}_{10}$. *Adv. Sci.*, **12**, 2410543 (2025).

Supplementary Fig. 6 XPS spectra of CTO at the pristine state, after discharging to 1.0 V, and after discharging to 0.01 V. **a** Evolution of Ce 3d high-resolution spectra. **b** Evolution of Ti 2p high-resolution spectra.

Comment 6: The specific capacity exhibits severe fluctuation at high current rates, please provide reasonable explanation for this.

Response: We thank the reviewer for raising this important point. To the best of our knowledge, similar capacity fluctuation phenomena during high-rate cycling have been observed in many other fast-charging anode materials (e.g., *Adv. Mater.*, 2020, 32, 1905295; *Adv. Mater.*, 2022, 34, 2107262; *Adv. Energy Mater.*, 2023, 13, 2302107; *Adv. Energy Mater.*, 2022, 12, 2102972). According to the explanations in prior studies (*Adv. Mater.*, 2024, 36, 2412266; *Nano Energy*, 2024, 119, 109065; *J. Energy Chem.*, 2022, 74, 34-44), the initial capacity increase is typically attributed to gradual electrolyte infiltration and the progressive activation of electrochemically accessible regions within the bulk electrode. Subsequent capacity decline is commonly associated with SEI growth, electrode passivation, or structural degradation.

In our case, multiple capacity fluctuations are observed during extended high-rate cycling. We speculate that it originates from the delayed activation of interior active

regions. Specifically, due to the micron-scale size of CTO particles and the large current densities applied, lithium diffusion during the early cycles is insufficient to fully access the particle interior. The rapid lithiation/delithiation at high rates limits lithium penetration depth, restricting full utilization of accessible sites. As cycling proceeds, the accumulated mechanical stress and repeated volume changes induce particle fragmentation or local SEI disruption, thereby exposing previously isolated regions. This reactivation process allows further utilization of internal intercalation sites, leading to repeated capacity recovery during prolonged cycling. Despite these capacity fluctuations, the CTO electrode maintains high capacity retention (> 80%) after thousands of cycles, demonstrating its excellent long-term cycling stability and practical potential for high-rate applications.

Comment 7: In Figure 3e, it is quite obvious that the potential has not reached an equilibrium state. Therefore, the accuracy of the diffusion coefficient calculation is questionable.

Response: We thank the reviewer for this valuable comment. To address this concern, we repeated the GITT measurements using an extended relaxation time of 90 minutes (instead of the original 20 minutes). This adjustment ensures that the potential sufficiently approaches a near-equilibrium state during each titration step, thereby improving the accuracy of the calculated diffusion coefficients.

The updated results confirm that CTO exhibits fast Li^+ diffusion kinetics, with average diffusion coefficients of $5.29 \times 10^{-11} \text{ cm}^2 \text{ s}^{-1}$ during lithiation and $9.39 \times 10^{-11} \text{ cm}^2 \text{ s}^{-1}$ during delithiation. These values are close to the previously calculated results, reaffirming the intrinsically rapid Li^+ transport properties of CTO.

Accordingly, we have updated **Fig. 3d-f** with the revised GITT profiles and diffusion coefficient plots. The updated average diffusion coefficients have also been incorporated into the revised manuscript and **Supplementary Table 2**.

Lines 294-297, Page 11: The average diffusion coefficients during lithiation and delithiation were found to be 5.29×10^{-11} and $9.39 \times 10^{-11} \text{ cm}^2 \text{ s}^{-1}$, respectively, which exceed those of many fast-charging electrode materials (**Supplementary Table**

2).

Fig. 3 Kinetic analysis for Li⁺ diffusion within CTO electrode. d GITT curves during lithiation and delithiation at a rate of 0.1 C. **e** Enlarged view of GITT curves for the determination of ΔE_{τ} and ΔE_s . **f** Calculated Li⁺ diffusion coefficients during both lithiation and delithiation processes.

Responses to the comments from Reviewer 2

Reviewer 2

General comments: I co-reviewed this manuscript with one of the reviewers who provided the listed reports. This is part of the *Nature Communications* initiative to facilitate training in peer review and to provide appropriate recognition for Early Career Researchers who co-review manuscripts.

Response: We sincerely thank you for your constructive feedback and valuable suggestions, as well as the time and effort you invested in the thorough evaluation of our manuscript.

Responses to the comments from Reviewer 3

Reviewer 3

General comments: In this work, the authors report an excellent electrochemical Li⁺ storage performance using the vacancy-ordered perovskite Ce_{2/3}TiO₃ as the anode materials for half and full cells at ambient temperature, compared to other perovskite and intercalation-type anodes. Especially, the moderate operating voltage (~0.8 V) avoids the safety hazard of graphite due to generating lithium dendrites. Moreover, the authors perform an exhaustive characterization for the crystal structure and charge-discharge process. The results are interesting and valuable. However, there exist some critical problems in the current version, especially about the charge-discharge mechanism. Therefore, I can't recommend its publication on Nature Communications unless the following issues are satisfactorily resolved.

Response: We sincerely thank the reviewer for the positive evaluation of our work. We greatly appreciate the reviewer's recognition of the study's novelty, the electrochemical advantages of CTO, and the comprehensive characterizations. At the same time, we fully acknowledge the reviewer's concerns regarding the charge-discharge mechanism and agree that these issues are critical for validating our conclusions.

In response, we have accordingly revised the manuscript to address all the comments. We hope that the point-by-point responses and newly added data satisfactorily address the reviewer's concerns. The detailed responses to each comment are provided below.

Comment 1: A-vacancy Ln_{2/3}TiO₃ is a type of perovskites that have been widely researched in other fields (Journal of Materials Science Letters, 1999, 18, 1181; Journal of Solid State Chemistry, 2000, 149, 354; J. Am. Ceram. Soc., 2002, 85 1339; Ceramics International, 2016, 42, 1698), including Ce_{2/3}TiO₃ (Journal of Alloys and Compounds, 2000, 305, 72). The authors should introduce the characters of Ln_{2/3}TiO₃ and the considerations that Ce_{2/3}TiO₃ is selected in this work.

Response: We sincerely thank the reviewer for this constructive suggestion.

A-site deficient perovskites with the general formula Ln_{2/3}TiO₃ (Ln = La, Ce, Pr,

Nd) adopt an ordered double-perovskite structure characterized by one-third A-site vacancies periodically arranged along the *c*-axis. This vacancy ordering stabilizes the TiO₆ octahedral framework and preserves long-range periodicity. Such structural features facilitate ion transport and accommodate reversible lattice changes during cycling. The intrinsic structural superiority renders A-site deficient perovskites promising for intercalation-type anode materials, which require both efficient ion mobility and robust structural stability. Although the electrical transport, dielectric, and magnetic properties of Ln_{2/3}TiO₃ have been widely studied, their potential for lithium-ion storage and the underlying microscopic mechanisms remain largely unexplored.

To the best of our knowledge, among Ln_{2/3}TiO₃ family, the electrochemical properties and associated lithium-storage mechanisms of Ce_{2/3}TiO₃ have never been previously reported. In contrast, derivative anodes based on other Ln_{2/3}TiO₃, such as Li_{0.27}La_{0.54}TiO_{2.945} (*Electrochem. Commun.*, 2013, 32, 5-8), La_{0.5}Li_{0.5}TiO₃ (*Nat. Commun.*, 2020, 11, 3490), Li_{0.35}Nd_{0.55}TiO₃ (*Nano Energy*, 2024, 119, 109065), and Li_{0.38}Pr_{0.54}TiO₃ (*Chem. Eng. J.*, 2024, 479, 147765), have been studied, but these anodes generally exhibit limited rate capability and insufficient long-term cycling stability compared to Ce_{2/3}TiO₃ reported in this study. Furthermore, previous studies on Ln_{2/3}TiO₃ anodes have largely focused on electrochemical property, with limited insight into the underlying structural mechanisms, leaving the structure-property relationship in A-site deficient perovskites insufficiently understood. Therefore, the selection of Ce_{2/3}TiO₃ in this study aims not only to achieve a breakthrough in electrochemical performance but also to elucidate a deeper understanding of structural mechanisms, thereby offering a new design paradigm for high-performance intercalation-type anode materials.

The relevant description and references have been added to the introduction section of the revised manuscript as follows:

Lines 86-98, Page 4: In this context, A-site deficient perovskites with the general formula Ln_{2/3}TiO₃ (Ln = La, Ce, Pr, Nd) represent an ideal structural prototype for intercalation-type anodes, intrinsically combining ordered cation vacancies with a robust TiO₆ octahedral framework^{39, 40, 41, 42, 43}. This vacancy ordering not only

modulates the Ti-O-Ti connectivity to promote Li⁺ transport but also mitigates volume changes during cycling to enhance structural stability, which are critical for high-performance anode materials. Several derivatives of this family, including Li_{0.27}La_{0.54}TiO_{2.945}, La_{0.5}Li_{0.5}TiO₃, Li_{0.35}Nd_{0.55}TiO₃, and Li_{0.38}Pr_{0.54}TiO₃, have been explored as the anodes for LIBs^{44, 45, 46, 47}. However, they usually suffer from limited rate capability and insufficient long-term cycling stability, while their underlying lithium-storage mechanisms at the microscopic level remain poorly understood. Therefore, exploring novel A-site deficient perovskite is essential to overcome current performance limitations and to establish universal structure-property relationships for high-performance intercalation-type anodes.

References:

39. Jung W. H. Structure, thermopower and electrical transport properties of La_{2/3}TiO_{3-δ}. *J. Mater. Sci. Lett.* **18**, 1181-1183 (1999).
40. Yoshii K. Synthesis and Magnetic Properties of Ln_{2/3}TiO₃ (Ln=Pr and Nd). *J. Solid State Chem.* **149**, 354-359 (2000).
41. Yoshioka H. Structure and Electrical Properties of A-site-Deficient Perovskite Compounds in the La_{2/3}TiO₃-La_{1/3}NbO₃ System. *J. Am. Ceram. Soc.* **85**, 1339-1342 (2002).
42. Bugrov A. N., *et al.* Soft chemistry synthesis and dielectric properties of A-site deficient perovskite-type compound La_{2/3}TiO_{3-δ}. *Ceram. Int.* **42**, 1698-1704 (2016).
43. Yoshii K. Structural and magnetic studies of the lanthanide deficient perovskite Ce_{2/3}TiO₃. *J. Alloys Compd.* **305**, 72-75 (2000).
44. Hua C., Fang X., Wang Z., Chen L. Lithium storage in perovskite lithium lanthanum titanate. *Electrochem. Commun.* **32**, 5-8 (2013).
45. Zhang L., *et al.* Lithium lanthanum titanate perovskite as an anode for lithium ion batteries. *Nat. Commun.* **11**, 3490 (2020).
46. Liu H., *et al.* Cation-deficient perovskite Li_{0.35}Nd_{0.55}TiO₃ as a high-performance anode for lithium-ion batteries. *Nano Energy* **119**, 109065 (2024).
47. Liu H., *et al.* A-site deficient perovskite lithium praseodymium titanate as a high-rate anode for lithium-ion batteries. *Chem. Eng. J.* **479**, 147765 (2024).

Comment 2: The authors emphasize the superiority of CTO with respect to the commercial anodes. Nowadays, a good low-temperature performance has become an urgent demand for commercial secondary batteries. In order to better satisfy the practical requirement, the low-temperature performance of CTO should be examined.

Response: We appreciate the reviewer's insightful suggestion.

In response, we have performed additional electrochemical measurements at 0 °C and -15 °C to evaluate the low-temperature performance of the CTO electrode. The results show that, although a capacity reduction is observed at low temperatures, CTO retains great rate capability and stable cycling performance. The low-temperature performance of CTO is better than that of conventional commercial anodes such as $\text{Li}_4\text{Ti}_5\text{O}_{12}$ and graphite, further highlighting its promising potential for practical applications.

The results have been added to **Supplementary Figs. 12 and 13**, and the relevant discussion has been included in the revised manuscript as follows:

Lines 237-248, Page 9: Low-temperature performance is also a critical metric for practical battery applications, so the electrochemical performance of CTO was evaluated at 0 and -15 °C, (**Supplementary Figs. 12 and 13**). At 0 °C, although a moderate reduction in capacity compared to room temperature, CTO still delivers a reversible capacity of 183 mAh g⁻¹ at 0.1 C. More notably, it maintains excellent rate capability, delivering 170, 161, 149, 130, 111, 89, and 54 mAh g⁻¹ as the rate increases from 0.5 C to 50 C. Long-term cycling tests further demonstrate outstanding stability, with negligible capacity decay over 1000 cycles at 10 C and over 10000 cycles at 20 C. Even at lower temperature of -15 °C, where Li⁺ diffusion kinetics are more severely limited, it can maintain considerable cycling stability at 5 C and 10 C. The low-temperature performance of CTO is better than conventional commercial anodes such as $\text{Li}_4\text{Ti}_5\text{O}_{12}$ and graphite^{58, 59, 60}, underscoring its promising potential for practical applications in extreme environments.

References:

58. Allen J. L., Jow T. R., Wolfenstine J. Low temperature performance of nanophase

$\text{Li}_4\text{Ti}_5\text{O}_{12}$. *J. Power Sources* **159**, 1340-1345 (2006).

59. Ho C.-K., Li C.-Y. V., Deng Z., Chan K.-Y., Yung H., Yang C. Hierarchical macropore-mesoporous shell carbon dispersed with $\text{Li}_4\text{Ti}_5\text{O}_{12}$ for excellent high rate sub-freezing Li-ion battery performance. *Carbon* **145**, 614-621 (2019).

60. Hou R., *et al.* An Unexpected Low-Temperature Battery Formation Technology Enabling Fast-Charging Graphite Anodes. *Adv. Funct. Mater.* **35**, 2500481 (2025).

Supplementary Fig. 12 Low-temperature electrochemical performance of CTO electrode at 0 °C a Charge-discharge profiles at various current rates from 0.1 C to 50

C. b Rate capability and corresponding coulombic efficiency at 0.1, 0.5, 1, 2, 5, 10, 20, and 50 C. **c** Cycling performance at 10 C. **d** Cycling performance at 20 C.

Supplementary Fig. 13 Low-temperature electrochemical performance of CTO electrode at -15 °C **a** Charge-discharge profiles at various current rates from 0.1 C to 20 C. **b** Rate capability and corresponding coulombic efficiency at 0.1, 0.5, 1, 2, 5, 10, and 20 C. **c** Cycling performance at 5 C. **d** Cycling performance at 10 C

Comment 3: The authors state that the intercalated Li⁺ occupy the A-site vacancies of

CTO in the whole lithiation process. However, I cannot fully agree with this view. $\text{Ce}_{2/3}\text{TiO}_3$ has the A-vacancy concentration of only 1/3. The resultant theoretical capacity is only ~47 mAh/g when all A vacancies are filled. Actually, it has been reported that the inserted Li^+ may occupy the O4 window positions in perovskite anodes (Chem Mater 2013, 25, 1607). If this case holds true in this work, it should be emphatically considered in the related experimental characterization and theoretical analysis.

Response: We sincerely thank the reviewer for this valuable comment. We fully agree that the relatively low A-site vacancy concentration (1/3 per formula unit) in $\text{Ce}_{2/3}\text{TiO}_3$ cannot account for the high observed capacity, and that additional lithium insertion sites beyond the A-site vacancies must be considered.

To address this concern, we have thoroughly revised the theoretical calculations. In the updated DFT calculations, we systematically identified all potential lithium insertion sites in both the vacancy-ordered tetragonal $\text{Ce}_{10}\text{Ti}_{16}\text{O}_{48}$ and the pseudo-cubic $\text{LiCe}_5\text{Ti}_8\text{O}_{24}$ structures (as shown in **Supplementary Figs. 31 and 33**). In addition to A-site vacancies, a series of energetically favorable interstitial sites were identified in the O4 windows between adjacent TiO_6 octahedra (i.e., sites $\text{Li}_4 - \text{Li}_7$ and $\text{Li}_{10} - \text{Li}_{12}$), consistent with the prior literature (e.g., *Chem. Mater.*, 2013, 25, 1607). The calculated formation energies for these O4 window sites are comparable to those of A-site vacancies (as shown in **Supplementary Tables 3 and 4**), indicating there are thermodynamically accessible and contribute significantly to the overall lithiation process.

To further confirm the theoretical predictions, we supplemented atomic-resolution integrated differential phase contrast scanning transmission electron microscopy (iDPC-STEM) on lithiated CTO, which enables simultaneous imaging of light and heavy elements with sub-angstrom resolution. The iDPC-STEM image reveals additional contrast at the O4 window sites, confirming lithium occupation at these interstitial positions (**Supplementary Fig. 32**).

Overall, both theoretical and experimental results demonstrate that Li^+ insertion in CTO occurs not only at A-site vacancies but also extensively at interstitial O4 window

sites between the TiO_6 octahedra, thereby providing a high lithium storage capacity.

Supplementary Fig. 31 Optimized structure models depicting potential Li^+ storage sites in the tetragonal phase.

Supplementary Fig. 32 iDPC-STEM image and corresponding structural model of lithiated CTO along the $[110]$ zone axis, highlighting Li^+ insertion into O4 windows between adjacent TiO_6 octahedra. The inserted Li^+ positions are indicated by red dashed

circles.

Supplementary Fig. 33 Optimized structure models depicting potential Li^+ storage sites in the pseudo-cubic phase.

Comment 4: In the pristine CTO sample, $\text{Ce}_{0.889}$ and $\text{Ce}_{0.444}$ layers uniformly alternate. And, the authors conclude that the intercalated Li^+ preferentially occupy the A-vacancies in the $\text{Ce}_{0.444}$ layers. So, the distribution of inserted Li^+ should be relatively uniform in all $\text{Ce}_{0.444}$ layers. And the lithiation-induced pseudo-cubic phase should homogeneously mix with the pristine tetragonal phase, as observed in Fig. 5g (middle 1.25 V). Why does a distinct boundary between the pure tetragonal and pure pseudo-cubic phases occur, as shown in Fig. 5b and Fig. S15? Moreover, the difference between blue and orange regions in Fig. S15 is hardly distinguished, how to determine the phase boundary?

Response: We appreciate the reviewer's thoughtful question regarding the phase boundary and the interpretation of Fig. S15. Our response is organized into two parts to address the reviewer's concerns in detail:

(1) Regarding the distinct phase boundary between tetragonal and pseudo-cubic phases:

In principle, Li^+ ions preferentially intercalate into the Ce-poor ($\text{Ce}_{0.444}$) layers, which could ideally result in a homogeneous mixture of the lithiation-induced pseudo-

cubic phase and the pristine tetragonal phase. However, under practical experimental conditions, lithium intercalation is rarely spatially uniform due to kinetic factors such as diffusion limitations, particle surface effects, and local structural variations. Given micron-scale size of CTO particles, lithiation typically initiates at the particle surface and propagates inward, leading to a spatially resolved phase-front propagation process. The pseudo-cubic phase nucleates at the particle surface and gradually extends into the bulk as lithiation proceeds. Additionally, the sample preparation by focus ion beam (FIB) slicing exposes the cross-section of particles, making the internal phase boundaries accessible for observation. As a result, a distinct phase boundary between the tetragonal and pseudo-cubic phases was observed at the intermediate state via STEM imaging. Analogous phase-front progression and two-phase coexistence behaviors have been widely observed in other electrodes (e.g., $\text{LiNi}_{0.8}\text{Co}_{0.15}\text{Al}_{0.05}\text{O}_2$, LiFePO_4 , Fe_3O_4 , TiO_2 , etc.), as revealed by *in situ* or *ex situ* characterizations (e.g., *ACS Energy Lett.*, 2020, 5, 2098 – 2105; *Science*, 2016, 353, 566-571; *ACS Appl. Energy Mater.*, 2020, 3, 8009 – 8016; *Nature Commun.*, 2016, 7, 11441; *Adv. Mater.*, 2017, 29, 1700236).

(2) *Regarding the visibility and identification of the phase boundary:*

We apologize for the confusion caused by the original HAADF-STEM image in Fig. S15. Although the phase regions were colored-labeled according to structural assignments, the contrast difference may appear subtle in grayscale rendering. To address this, we reconstructed a color-coded atomic contrast heatmap from the raw HAADF-STEM image (now shown in **Supplementary Fig. 24**). This method maps atomic column intensities to color gradients, which clearly visualizes the periodic Z-contrast difference between Ce-rich and Ce-poor layers in the tetragonal phase. In contrast, this contrast variation is significantly diminished in the pseudo-cubic region, enabling clear identification of the phase boundary.

Supplementary Fig. 24 **a** HAADF-STEM image along the [100] zone axis of CTO after long-term cycling. The boundary between tetragonal and pseudo-cubic phases is indicated by white dash lines. **b** Reconstructed atomic contrast heatmap derived from the raw HAADF-STEM image, highlighting the coexistence of tetragonal and pseudo-cubic phases. **c** Line intensity profile along the blue line in the tetragonal region, showing pronounced periodic modulation of Ce atomic column intensities. **d** Line intensity profile along the orange line in the pseudo-cubic phase, where the contrast variation of Ce atomic columns is largely diminished.

Comment 5: The authors state that the valence configurations of Ti atoms are $3d^04s^0$ in pristine structure, and $3d^14s^0$ in lithiated structures. If so, why are remarkably asymmetric for the spin-up and spin-down PDOS of Ti-d and Ti-p in pristine structure (Fig. S22)? And why are symmetric or quasi-symmetric for the spin-resolved PDOS of Ti $3d^14s^0$ under lithiations?

Response: We thank the reviewer for pointing out this important issue. We sincerely apologize for the confusion caused by the previously presented PDOS results, which arose from the inappropriate structural models in the initial calculations.

To address this issue, we reconstructed two representative structural models that

more accurately reflect the experimentally observed phases: (1) a tetragonal vacancy-ordered $\text{Ce}_{10}\text{Ti}_{16}\text{O}_{48}$ superlattice representing the pristine state, and (2) a pseudo-cubic $\text{LiCe}_5\text{Ti}_8\text{O}_{24}$ structure corresponding to the topologically transformed phase upon partial lithiation. Based on these refined models, we recalculated the spin-resolved PDOS of Ti d orbitals at representative lithiation states. In the pristine state, the PDOS exhibits symmetric spin-up and spin-down distributions, consistent with the Ti^{4+} ($3d^0$) electronic configuration. Upon partial lithiation, the pseudo-cubic phase shows a slight spin asymmetry of Ti d states, indicating partial electron occupation of Ti $3d$ orbitals. After full lithiation, the spin asymmetry becomes more pronounced, suggesting further electron population into Ti $3d$ orbitals. These results confirm a multistep Ti^{4+} reduction process during lithiation.

The updated PDOS results are now in **Supplementary Fig. 34**, and the relevant discussion has been updated in the revised manuscript as follows:

Lines 458-461, Page 17: The Projected DOS of Ti d orbitals (**Supplementary Fig. 34**) reveal electron transfer from lithium to Ti $3d$ orbitals, resulting in a progressive transition from spin-symmetric to spin-asymmetric Ti $3d$ occupancy, indicative of the stepwise reduction of Ti^{4+} .

Supplementary Fig. 34 Projected DOS of Ti *d* orbitals, illustrating electronic structure evolution upon lithiation. **a** Pristine tetragonal phase. **b** Partial lithiation-induced pseudo-cubic phase. **c** Fully lithiated CTO.

Comment 6: An obviously contradictory result. The DOS results of pristine model are inconsistent in Fig. 6b and Fig. 22. For example, the former shows a wide bandgap about 2 eV, whereas the latter shows the nature of a half metal.

Response: We thank the reviewer for highlighting this issue and apologize for the confusion caused by the previous calculation results.

As noted in our response to **Comment 5**, we have revised the DFT calculations by reconstructing more accurate structural models: a tetragonal $\text{Ce}_{10}\text{Ti}_{16}\text{O}_{48}$ for the pristine state and a pseudo-cubic $\text{LiCe}_5\text{Ti}_8\text{O}_{24}$ for the topologically transformed phase. The recalculated projected DOS reveals that the pristine state exhibits a wide bandgap of

~1.8 eV. Upon lithiation, the fermi level gradually shifts toward the conduction band, and electronic states emerge near the fermi level, indicating a transition toward half-metallic behavior. These trends are consistent with the evolution of the spin-resolved Ti *d* PDOS (now presented in **Supplementary Fig. 34**).

Accordingly, we have revised **Fig. 6b** and updated the corresponding discussion in the revised manuscript as follows:

Lines 454-461, Page 17: Concurrently, the electronic structure undergoes significant modifications during lithiation, as evidenced by the projected density of states (DOS). The pristine tetragonal phase exhibits a wide bandgap of ~1.8 eV, with the conduction band far from the fermi level (**Fig. 6b**). Upon Li⁺ insertion, the bandgap gradually narrows and the electronic states near the fermi level increase. The projected DOS of Ti *d* orbitals (**Supplementary Fig. 34**) reveal electron transfer from lithium to Ti *3d* orbitals, resulting in a progressive transition from spin-symmetric to spin-asymmetric Ti *3d* occupancy, indicative of the stepwise reduction of Ti⁴⁺.

Fig. 6 b Projected DOS comparison for three representative states: pristine tetragonal phase, intermediate pseudo-cubic phase, and fully lithiated phase.

Comment 7: The constructed pristine CTO model is Ce₃Ti₈O₂₄ in Fig. 6a, not the Ce₄Ti₈O₂₄ described in DFT calculations, Methods. If the constructed supercell model is Ce₃Ti₈O₂₄, the A-vacancy concentration is up to 5/8, much higher than the experimental 1/3 in Ce_{2/3}TiO₃. In addition, the constructed pristine model is a quasi-cubic supercell, different from the synthesized tetragonal cell ($c \approx 2a$). How do authors ensure the reliability of DFT calculations?

Response: We thank the reviewer for this insightful comment and sincerely apologize for the inconsistency between the previously constructed DFT model and the experimental structure.

To address this concern, we have reconstructed a more accurate structural model for pristine $\text{Ce}_{2/3}\text{TiO}_3$, adopting a vacancy-ordered tetragonal superlattice $\text{Ce}_{10}\text{Ti}_{16}\text{O}_{48}$. This model features an A-site vacancy concentration of $3/8$, which closely approximates the experimental value of $1/3$ while maintains a balance between computational complexity and accuracy. This model also retains the experimentally observed tetragonal symmetry with $c \approx 2a$, ensuring consistency with the actual crystallographic phase. To reflect the lithiation-induced structural evolution, we further constructed a pseudo-cubic $\text{LiCe}_5\text{Ti}_8\text{O}_{24}$ model representing the topologically transformed phase after partial lithiation. These two models ($\text{Ce}_{10}\text{Ti}_{16}\text{O}_{48}$ and $\text{LiCe}_5\text{Ti}_8\text{O}_{24}$) were systematically employed to calculate lithium insertion energetics, electronic structure, and Li^+ diffusion barriers. These modifications significantly improve the reliability and accuracy of our DFT calculations, ensuring closer alignment with the experimental results.

We have thoroughly updated **Fig. 6** and the corresponding discussions (**Lines 428–485, Pages 16–18**) in the revised manuscript, and have also updated the relevant figures (**Supplementary Figs. 31-36**) and data tables (**Supplementary Tables 3 and 4**) in Supplementary Information.

Fig. 6 DFT calculations for Li^+ intercalation dynamics and phase evolution. a Geometric structures illustrating Li^+ insertion sites and associated structural evolution during lithiation. **b** Projected DOS comparison for three representative states: pristine tetragonal phase, intermediate pseudo-cubic phase, and fully lithiated phase. **c** Charge density distribution maps projected along the (100) plane for the tetragonal and pseudo-cubic phases. **d** BVS energy maps and predicted Li^+ diffusion pathways in tetragonal and pseudo-cubic structures. **e** Comparison of Li^+ diffusion energy barriers in tetragonal and pseudo-cubic structures.

Comment 8: In Fig S8, the full cell has a coulombic efficiency only ~30% for the first cycle, much lower than the 64.4% of half cell, why?

Response: We sincerely thank the reviewer for pointing out this important issue and apologize for the oversight regarding the low first-cycle coulombic efficiency of the full cell.

After careful evaluation, we think that the low first-cycle coulombic efficiency did not originate from the intrinsic properties of the electrodes, but was primarily caused by suboptimal capacity balancing in the previous full cell configuration. Specifically, the previous full cell suffered from an inappropriate capacity ratio between the cathode and anode, leading to excessive irreversible lithium consumption during the first cycle. This capacity mismatch hindered efficient utilization of the active lithium source, resulting in a low first-cycle coulombic efficiency.

To address this issue, we reassembled the full cell with optimized electrode mass loadings to ensure a more appropriate N/P ratio of 1.1-1.2. As a result, the first-cycle coulombic efficiency was significantly improved to 62.5%, which was close to that of the half cell (64.4%). Moreover, the optimized full cell exhibited enhanced rate capability and cycling performance. It delivered a capacity retention of 61.3% at 10 C and maintained 80% capacity retention after 1000 cycles at 5 C, demonstrating the excellent electrochemical performance of the full cell.

The updated electrochemical results of the full cell have been incorporated into **Supplementary Fig. 15**, and the corresponding discussion has been added to the revised manuscript as follows:

Lines 255-262, Page 10: At 0.1 C, the full cell delivers a high discharge capacity of 150 mAh g⁻¹, with a first-cycle coulombic efficiency of 62.5% (**Supplementary Fig. 15a**), comparable to that of the half cell. As the rate increases to 0.5, 1, 2, 5, and 10 C, the full cell maintains specific capacities of 125, 115, 108, 100, and 92 mAh g⁻¹, respectively (**Supplementary Fig. 15b**). Notably, the capacity retention at 10 C reaches 61.3%, demonstrating excellent rate capability. Furthermore, long-term cycling at 5 C confirms outstanding stability, with 80% capacity retention after 1000 cycles (**Supplementary Fig. 15c**).

Supplementary Fig. 15 Electrochemical performance of LiFePO₄||CTO full cells. **a** Charge/discharge curves of the first cycle, showing an initial coulombic efficiency of 62.5%. **b** Charge/discharge profiles at different current rates from 0.1 C to 10 C. **c** Rate capability and long-term cycling performance at 5 C.

Comment 9: Line 313, “the valance reduction” may be “the valence reduction”.

Response: We thank the reviewer for pointing out this typographical error. The term “valance” has been corrected to “valence” in the revised manuscript (**Line 379, Page 14**).

Responses to the comments from Reviewer 4

Reviewer 4

General comments: In this manuscript, vacancy-ordered $\text{Ce}_{2/3}\text{TiO}_3$ (CTO) is reported as a perovskite-type anode with high gravimetric capacity ($> 200 \text{ mAh g}^{-1}$ at $\approx 0.8 \text{ V}$), ultrafast-rate capability ($\approx 80 \text{ mA h g}^{-1}$ at 50 C), and sustained cycling stability (90% capacity retention after 10,000 cycles). Accelerating Li^+ transport by A-site vacancy ordering would be a new route which, if confirmed under electrode loadings, can be much better than graphite and spinel $\text{Li}_4\text{Ti}_5\text{O}_{12}$. The proof of concept is achieved utilizing significant measurements, however need to refine some sections as: electrochemical access is restricted to thin ($\approx 1.5 \text{ mg cm}^{-2}$) electrodes without any volumetric benchmarking; the 10,000 cycle claim rests simply upon a single end-point value; and the "pseudo-cubic" lithiated phase, as well as potential Ce-valence changes. Consequently, I recommend minor revision with an emphasis of the need for extensive electrode level validation, detailed cycle dataset, quantitative structural analysis and direct comparison with the industry benchmarks.

Response: We sincerely thank the reviewer for the constructive and insightful comments. We appreciate the reviewer's recognition of the novelty and potential of the vacancy-ordered CTO as a high-performance anode. In response to the reviewer's comments, we have accordingly revised the manuscript and provided additional experimental results as outlined below:

(1) Electrode-level validation and industry benchmarking:

To address this comment, we conducted additional electrochemical measurements using thick CTO electrodes with a high mass loading of $\sim 10.2 \text{ mg cm}^{-2}$. The results (now presented in **Supplementary Fig. 10**) demonstrate that the thick CTO electrode retains excellent rate capability (e.g., 67.7% capacity retention at 20 C) and cycling stability (80% capacity retention after 1000 cycles at 20 C). We also evaluated the volumetric capacity of both thin ($\sim 1.5 \text{ mg cm}^{-2}$) and thick electrodes, which achieve maximum volumetric capacities of 550 mAh cm^{-3} and 394 mAh cm^{-3} , respectively. For comparison, we benchmarked CTO against representative micro-sized and nano-structured intercalation-type anodes (**Supplementary Fig. 11**), confirming its superior

performance over conventional anode materials. These results highlight the practical applicability of CTO. The corresponding discussion has been added to the revised manuscript as follows:

Lines 222-236, Page 9: Most fast-charging anode materials reported to date rely on thin electrodes with low mass loadings ($1\text{-}2\text{ mg cm}^{-2}$), which limit their practical applicability. To further demonstrate the superior electrochemical performance of CTO under application-relevant conditions, we evaluated thick electrodes with a high mass loading of 10.2 mg cm^{-2} . As shown in **Supplementary Fig. 10**, the CTO electrode delivers specific capacities of 193, 173, 160, 146, 117, 89, 63, and 30 mAh g^{-1} at current rates from 0.1 C to 50 C, demonstrating excellent rate capability even at a high loading. Moreover, it retains 80% of the initial capacity after 1000 cycles at 20 C, maintaining considerable cycling stability. Beyond gravimetric performance, we also assessed the volumetric capacity of CTO. At a low mass loading of 1.5 mg cm^{-2} , CTO exhibits a maximum volumetric capacity of 550 mAh cm^{-3} , and still achieves 394 mAh cm^{-3} at the high loading of 10.2 mg cm^{-2} (**Supplementary Fig. 11**). This performance significantly outperforms that of previously reported micro-sized or nano-structured intercalation-type anodes (**Supplementary Fig. 11c**), further highlighting the practical advantages of CTO for high-energy-density and fast-charging LIBs.

Supplementary Fig. 10 Electrochemical performance of CTO electrode with a high mass loading of 10.2 mg cm⁻². **a** Charge-discharge profiles at various current rates from 0.1 C to 50 C. **b** Rate capability and corresponding coulombic efficiency at 0.1, 0.5, 1, 2, 5, 10, 20, and 50 C. **c** Cycling performance at a high rate of 20 C.

Supplementary Fig. 11 Volumetric performance of CTO electrode with low (1.5 mg cm⁻²) and high (10.2 mg cm⁻²) mass loadings. **a** Cross-sectional SEM images. **b** Areal current density versus volumetric capacity. **c** Comparison of volumetric capacities at 1 C and 10 C with previously reported intercalation-type anodes^{11, 20, 21, 22, 23, 24, 25, 26, 27, 28, 29, 30, 31}.

(2) Clarification of the 10000-cycle dataset:

In response, we have provided representative charge-discharge profiles recorded throughout the 10000-cycle test at 20 C (now presented in **Supplementary Fig. 7**), clearly demonstrating exceptional cycling stability with minimal capacity decay

between the initial and final cycles.

Supplementary Fig. 7 Detailed discharge/charge curves during prolonged high-rate cycling at 20 C.

(3) *Structural validation of the pseudo-cubic phase:*

To further substantiate the lithiation-induced transition from tetragonal to pseudo-cubic phase, we have supplemented additional experimental evidence and analysis.

Specifically, pair distribution function (PDF) refinements of pristine and lithiated samples further confirm a transition from a long-range ordered tetragonal structure to a short-range ordered pseudo-cubic phase upon lithiation (**Supplementary Fig. 29**). Moreover, *ex situ* Raman spectroscopy collected near the transition voltage of 1.25 V reveals progressive shifts in the dominant vibrational modes and merging of low-frequency modes as lithiation proceeds, consistent with an increase in structural symmetry during phase transition (**Supplementary Fig. 19**). The details are provided in our responses to **Comments 3** and **4** below.

(4) *Ce valence state evolution:*

The Ce M_{4,5}-edge EELS spectra (**Supplementary Fig. 26**) confirm that the valence state of Ce remains unchanged during lithiation. To further validate this, we conducted *ex situ* Ce 3d XPS measurements at different states (**Supplementary Fig. 6**),

which consistently reveal the electrochemical inactivity of Ce throughout the lithiation process. These complementary results further indicate that the redox activity during lithiation originates from Ti rather than Ce, as further elaborated in our response to **Comment 1** below.

Comment 1: First, the reversible capacity of 220–221 mA h g⁻¹ at 0.1 C surpasses the Ti⁴⁺ → Ti³⁺ theoretical limit for CTO (≈ 142 mA h g⁻¹), while EELS shows no Ce valence shift, implying that the extra ≈ 80 mA h g⁻¹ likely arises from over-estimated active mass or parasitic side reactions rather than intrinsic redox.

Response: We thank the reviewer for this insightful comment. The concern regarding the reversible capacity exceeding the theoretical Ti⁴⁺ → Ti³⁺ redox limit (≈ 142 mAh g⁻¹) is valid under the assumption of a single-electron redox. However, multiple lines of evidence support the occurrence of a two-electron Ti⁴⁺ → Ti²⁺ redox process during lithiation, corresponding to a theoretical capacity of 284 mAh g⁻¹, which sufficiently accounts for the measured capacity of 221 mAh g⁻¹.

Specifically:

1. The CV curves (**Fig. 2a**) exhibit three distinct redox couples at 1.45 V/1.24 V, 0.81 V/0.66 V, and 0.30 V/0.10 V. The redox couples at 1.45 V/1.24 V and 0.81 V/0.66 V are assigned to the stepwise Ti⁴⁺/Ti³⁺ redox process, consistent with previous reports on Ti-based anodes (e.g., *Nano Energy*, 2024, 119, 109065; *Chem. Eng. J.*, 2024, 479, 147765; *Electrochem. Commun.*, 2013, 32, 5-8; *Adv. Energy Mater.*, 2023, 13, 2302015). The redox couple at lower potential 0.30 V/0.10 V is attributed to a subsequent Ti³⁺/Ti²⁺ redox transition under deep lithiation, which is commonly observed below 0.5 V in other Ti-based anodes (e.g., *Energy Environ. Sci.*, 2017, 10, 1456-1464; *Adv. Sci.*, 2025, 2410543; *Nano Energy*, 2022, 94, 106972; *ACS Appl. Mater. Interfaces*, 2024, 16, 898 – 906).
2. Ti L_{2,3}-edge EELS spectra (**Supplementary Fig. 25**) collected at the pristine state, 1.25 V, and 0.01 V show progressive shifts of the L₃ and L₂ peaks toward lower energies, indicating the stepwise reduction of Ti from +4 to lower oxidation states. In contrast, Ce M_{4,5}-edge EELS spectra remain unchanged throughout lithiation,

confirming that Ce does not participate in redox reactions.

- To further corroborate our interpretation, we have supplemented *ex situ* XPS measurements on CTO (**Supplementary Fig. 6**). The Ti *2p* XPS peaks exhibit a continuous shift toward lower binding energies during lithiation, consistent with the $\text{Ti}^{4+} \rightarrow \text{Ti}^{3+} \rightarrow \text{Ti}^{2+}$ redox sequence. Meanwhile, the Ce *3d* spectra remain unchanged, further confirming the electrochemical inactivity of Ce during lithiation.

Overall, these results consistently demonstrate the occurrence of a $\text{Ti}^{4+} \rightarrow \text{Ti}^{2+}$ two-electron redox process in CTO and rule out the Ce redox, thereby accounting for the observed high reversible capacity.

Supplementary Fig. 6 XPS spectra of CTO at the pristine state, after discharging to 1.0 V, and after discharging to 0.01 V. **a** Evolution of Ce *3d* high-resolution spectra. **b** Evolution of Ti *2p* high-resolution spectra.

Comment 2: Second, the initial Coulombic efficiency is only 64 %, yet the solid-electrolyte interphase is uncharacterised, leaving the source of irreversible capacity loss- and its consequences for long-term stability—unclear.

Response: We thank the reviewer for highlighting this important point regarding solid electrolyte interphase (SEI) characterization. In response, we conducted additional SEI characterizations on CTO electrodes at different stages of the initial discharge (pristine, discharging to 0.5 V, and 0.01 V) using transmission electron microscopy (TEM) and X-ray photoelectron spectroscopy (XPS).

As shown in **Supplementary Fig. 4**, TEM images reveal a progressive thickening of the surface passivation layer during initial lithiation, indicating the formation and growth of SEI. Corresponding Li *1s* XPS spectra show the emergence and evolution of characteristic SEI components, including LiF, Li₂CO₃, and ROCO₂Li. As lithiation proceeds, the relative intensities of LiF and Li₂CO₃ signals increase while those of organic species decrease, suggesting an enrichment of inorganic SEI components. The irreversible lithium consumption associated with SEI formation accounts for the initial coulombic efficiency of ~64%. Meanwhile, the formation of a robust, LiF/Li₂CO₃-rich inorganic SEI layer is favorable for improving mechanical and interfacial stability of the electrodes, thereby contributing to the long-term cycling stability.

These results have been added to **Supplementary Fig. 4**, and the corresponding discussion has been included in the revised manuscript as follows:

Lines 171-181, Page 7: The initial capacity loss is primarily attributed to the formation of the solid electrolyte interphase (SEI). As shown in **Supplementary Fig. 4**, TEM images reveal progressive thickening of the SEI layer on CTO particle surfaces during the first discharge. High-resolution Li *1s* X-ray photoelectron spectroscopy (XPS) spectra indicate that the SEI consists of both organic (ROCO₂Li) and inorganic (Li₂CO₃ and LiF) components, which are from different interfacial reactions. ROCO₂Li and Li₂CO₃ are generated through the reductive decomposition of carbonate solvents and side reactions with conductive carbon, while LiF arises from the decomposition of LiPF₆ salt. As lithiation proceeds, the relative intensities of Li₂CO₃ and LiF signals increase while those of organic species decrease, suggesting a gradual enrichment of inorganic SEI components at low potentials, which leads to irreversible lithium consumption.

Supplementary Fig. 4 Evolution of SEI layer on CTO during initial lithiation. **a-c** TEM images of CTO particle surfaces at the pristine state (**a**), after discharging to 0.5 V (**b**), and 0.01 V (**c**), showing the formation and growth of the SEI layer. **d-f** Corresponding high-resolution Li *1s* XPS spectra, revealing the chemical evolution of Li-containing species during lithiation.

Comment 3: Third, the purported reversible transition from tetragonal to pseudo-cubic symmetry is supported only by in-situ XRD peak shifts and qualitative STEM images; rigorous Rietveld refinement of the lithiated phase and Ce-valence analysis are required to validate this structural evolution.

Response: We sincerely thank the reviewer for this constructive suggestion. To further substantiate the structural transition and clarify the Ce valence evolution, we have conducted additional structural and spectroscopic analysis, as detailed below:

(1) *Structural validation via pair distribution function (PDF) analysis:*

To provide more rigorous structural information beyond XRD and STEM, we conducted PDF refinements on both pristine and lithiated CTO samples. The results confirm that the pristine and lithiated states correspond to a long-range ordered tetragonal phase and a short-range ordered pseudo-cubic phase, respectively.

Additionally, in the short-range region (3.1 - 6.1 Å), the lithiated sample exhibits a new Ce-O correlation peak at ~5.0 Å and enhanced intensities of Ce1-Ti and Ce1-Ce2 correlations. These changes indicate a rearrangement of the local Ce coordination environment upon lithiation. Moreover, the Ce1-Ce1, Ce-O, and Ce1-Ce2 peaks in the lithiated state exhibit narrower widths compared to those in the pristine state, suggesting improved short-range ordering of Ce atoms. The PDF results provide additional evidence for a lithiation-induced transition in both structural symmetry and local ordering.

The PDF results have been added to **Supplementary Fig. 29**, and the corresponding discussion has been included in the revised manuscript as follows:

Lines 402-414, Page 15: Pair distribution function (PDF) analysis was performed to compare the pristine and lithiated states. As shown in **Supplementary Fig. 29a**, the PDF refinements reveal a transition from long-range tetragonal ordering to short-range pseudo-cubic ordering, consistent with the structural models in **Supplementary Fig. 29b**. In the short-range region (3.1 - 6.1 Å), the intensities of Ce1-Ti and Ce1-Ce2 correlations increase, and a new Ce-O correlation emerges at ~5.0 Å upon lithiation (**Supplementary Fig. 29c**), indicating a rearrangement of the Ce coordination environment. Moreover, the Ce1-Ce1, Ce-O, and Ce1-Ce2 peaks become noticeably narrower after lithiation (**Supplementary Fig. 29d**), reflecting enhanced short-range ordering of Ce atoms. These results collectively demonstrate that lithiation triggers a topological transition characterized by enhanced short-range structural ordering around Ce coordination environment.

Supplementary Fig. 29 PDF analysis revealing lithiation-induced transformation from long-range tetragonal ordering to short-range pseudo-cubic ordering. **a** PDF refinements of pristine and lithiated CTO. **b** Structural models of the long-range ordered tetragonal and short-range ordered pseudo-cubic phases. **c** Comparison of PDF peaks in the short-range region (3.1 - 6.1 Å), where the three dominant peaks are assigned to Ce1-Ti1 (~3.4 Å), Ce1-Ce1 (~3.8 Å), and Ce1-Ce2 (~5.5 Å) correlations, additional minor peaks are attributed to Ti-O and Ce-O distances. **d** Comparison of Ce1-Ce1, Ce-O, and Ce1-Ce2 peaks.

(2) *Ce valence evolution confirmed by ex situ XPS:*

As detailed in our response to **Comment 1**, we have conducted *ex situ* Ce 3d XPS measurements on CTO at the pristine state, a partially lithiated state (1.0 V), and a fully lithiated state (0.01 V). The Ce 3d XPS spectra remain essentially unchanged across these states, indicating that Ce maintains a stable +3 oxidation state throughout the lithiation process. This result is consistent with the Ce M_{4,5}-edge EELS spectra presented in **Supplementary Fig. 26** and confirms the electrochemical inactivity of Ce. These findings reveal that the structural transition is induced by lithium insertion and

associated topological rearrangement, rather than changes in Ce valence. The *ex situ* Ce 3d XPS spectra have been included in **Supplementary Fig. 6**.

Comment 4: Finally, complementary diagnostics--such as operando or ex-situ Raman spectroscopy near the 1.25 V transition and post-cycling XPS or FTIR of the electrodes--are needed to corroborate the symmetry change and identify SEI constituents responsible for the low first-cycle efficiency.

Response: We thank the reviewer for this valuable suggestion. In response, we have performed additional characterizations to corroborate the lithiation-induced symmetry change near the 1.25 V and to identify the SEI constituents after cycling.

(1) Ex situ Raman spectroscopy near the 1.25 V:

To capture the symmetry changes across the phase transition region, we have conducted *ex situ* Raman spectroscopy on CTO electrodes at a series of voltages (1.35 V, 1.30 V, 1.25 V, 1.20 V, and 1.10 V). As the voltage decreases, the Raman spectra exhibit evident changes, including a progressive downshift of the main peaks, disappearance of shoulder modes at ~ 124 and ~ 200 cm^{-1} , and increased intensities of vibrational modes at ~ 242 and ~ 664 cm^{-1} . These spectral changes, especially the gradual merging of low-frequency modes in the 100–200 cm^{-1} region, indicate an increase in structural symmetry, consistent with the transition from a tetragonal phase to a higher-symmetry pseudo-cubic phase.

The Raman spectra have been added to **Supplementary Fig. 19**, and the relevant discussion has been included in the revised manuscript as follows:

Lines 320-325, Page 12: Raman spectra collected across the phase transition region (**Supplementary Fig. 19**) reveal systematic vibrational changes during lithiation. The gradual downshift of main peaks, along with the progressive merging of low-frequency modes, indicates an increased structural symmetry, consistent with the transition toward a pseudo-cubic phase with higher symmetry.

Supplementary Fig. 19 *Ex situ* Raman spectra of CTO across the phase transition region. Raman spectra were collected at a series of discharge voltages (1.35 V, 1.30 V, 1.25 V, 1.20 V, and 1.10 V) to monitor structural evolution during lithiation. As the discharge proceeds, the main peaks at ~ 172 and ~ 354 cm^{-1} gradually shift to lower wavenumbers, while the shoulder peaks at ~ 124 and ~ 200 cm^{-1} progressively weaken and merge. Meanwhile, the intensities of modes at ~ 242 and 664 cm^{-1} increase below 1.30 V. These spectral changes, particularly the progressive merging of low-frequency modes in the $100\text{-}200$ cm^{-1} region, indicate a gradual increase in structural symmetry, consistent with the transition from a tetragonal to a higher-symmetry pseudo-cubic phase.

(2) *Post-cycling XPS analysis of SEI constituents:*

As detailed in our response to **Comment 2**, we have already provided TEM images and Li *1s* XPS spectra of CTO electrodes at different states of the initial discharge, which elucidates the origin of the low first-cycle coulombic efficiency. To further address the reviewer's concern, we have now included additional TEM and Li *1s* XPS analysis on the CTO electrode after cycling. The post-cycling Li *1s* XPS spectrum reveals that the SEI is predominately composed of inorganic components such as LiF and Li_2CO_3 , along with a minor organic component ROCO_2Li , which are nearly consistent with the SEI formed during the initial discharge. TEM analysis shows a slight increase in SEI thickness after cycling, indicating gradual but stable SEI evolution.

These results confirm that the SEI layer remains compositionally stable and mechanically robust after cycling, which enables the interfacial stability and long-term durability of the electrode.

The post-cycling TEM and XPS results have been added to **Supplementary Fig. 5**, and the relevant discussion has been included in the revised manuscript as follows:

Lines 182-186, Page 7: After cycling, the SEI maintains a similar chemical composition but becomes slightly thicker, with further accumulation of inorganic components such as LiF and Li₂CO₃ (**Supplementary Fig. 5**). This robust inorganic-rich SEI enhances the mechanical and chemical stability of the electrode interface, which is critical for mitigating continuous electrolyte decomposition and enabling long-term cycling durability.

Supplementary Fig. 5 a TEM image of the CTO electrode after cycling. b Corresponding high-resolution Li 1s XPS spectrum.

Response Letter to Reviewers

Responses to the comments from Reviewer 1

Comments: The authors have addressed the questions in the revised manuscript. It is thus recommended to be accepted.

Response: We sincerely thank the reviewer for the positive evaluation and recommendation. We are delighted that our revisions have satisfactorily addressed the concerns raised.

Responses to the comments from Reviewer 2

Comments: I co-reviewed this manuscript with one of the reviewers who provided the listed reports. This is part of the Nature Communications initiative to facilitate training in peer review and to provide appropriate recognition for Early Career Researchers who co-review manuscripts.

Response: We sincerely thank the reviewer for the valuable comment and constructive feedback.

Responses to the comments from Reviewer 3

Comments: The authors have provided thorough and detailed responses and revisions to my comments, which appear to have addressed all of my concerns. One minor suggestion is that the solid circles representing capacity and Coulombic efficiency in all cycling plots should be clearly labeled. Otherwise, it would be difficult for beginners in battery research to distinguish which corresponds to capacity and which to Coulombic efficiency. Overall, I recommend the publication of this paper on Nature Communications.

Response: We sincerely thank the reviewer for the positive evaluation and recommendation for publication. We also appreciate the constructive suggestion regarding the labeling of solid circles in the cycling plots. Following the advice, we have carefully revised all relevant figures to clearly distinguish capacity and Coulombic efficiency, thereby improving clarity for readers.

Responses to the comments from Reviewer 4

Comments: I have reviewed the manuscript and I find that the authors have complied with all comments and criticisms outlined in my earlier review of the manuscript. The revised manuscript is therefore acceptable as it is.

Response: We sincerely thank the reviewer for the positive comment. We are grateful for the constructive feedback provided throughout the review process.